# Afferent specific role of NMDA receptors for the circuit integration of hippocampal neurogliaform cells

R. Chittajallu[1], J.C. Wester[1], M.T. Craig [1,2], E. Barksdale[1], X.Q. Yuan[1], G. Akgül[1], C. Fang[1], D. Collins [1], S. Hunt[1], K.A. Pelkey[1] & C.J. McBain [1]

Appropriate integration of GABAergic interneurons into nascent cortical circuits is critical for ensuring normal information processing within the brain. Network and cognitive deficits associated with neurological disorders, such as schizophrenia, that result from NMDA receptor-hypofunction have been mainly attributed to dysfunction of parvalbumin-expressing interneurons that paradoxically express low levels of synaptic NMDA receptors. Here, we reveal that throughout postnatal development, thalamic, and entorhinal cortical inputs onto hippocampal neurogliaform cells are characterized by a large NMDA receptor-mediated component. This NMDA receptor-signaling is prerequisite for developmental programs ultimately responsible for the appropriate long-range AMPAR-mediated recruitment of neurogliaform cells. In contrast, AMPAR-mediated input at local Schaffer-collateral synapses on neurogliaform cells remains normal following NMDA receptor-ablation. These afferent specific deficits potentially impact neurogliaform cell mediated inhibition within the hippocampus and our findings reveal circuit loci implicating this relatively understudied interneuron subtype in the etiology of neurodevelopmental disorders characterized by NMDA receptor-hypofunction.

---

[1] Section on Cellular and Synaptic Physiology, Eunice Kennedy–Shriver National Institute of Child Health and Human Development, National Institutes of Health, Bethesda, Maryland 20892, USA. [2] Present address: Institute of Biomedical and Clinical Sciences, University of Exeter Medical School, Hatherly Laboratories, University of Exeter, Exeter EX4 4PS, UK. Correspondence and requests for materials should be addressed to C.J.M. (email: mcbainc@mail.nih.gov)

Normal brain function relies on the appropriate interaction between excitation (E) and inhibition (I) in the CNS, the latter mediated via the recruitment of ~20 subtypes of GABAergic interneurons (INs) in the CA1 hippocampus alone[1, 2]. IN dysfunction, resulting in E/I imbalance, is implicated in a number of neurodevelopmental disorders including epilepsy, schizophrenia and autism[3, 4]. In particular, hypofunction of NMDA receptor (NMDAR) mediated-signaling onto INs has emerged as a cause of various neurological disorders[5, 6]. However, despite this link little is known regarding the integration of IN subtypes at the basic synaptic physiological level following genetic ablation of NMDARs during embryonic and postnatal development[7, 8].

Recent technological developments that permit targeting of specific IN cohorts[9] are allowing for systematic dissection of their respective genetic programs and physiological roles. For example, optogenetic manipulation of parvalbumin (PV)-expressing IN activity, a subpopulation of which impart perisomatic inhibition onto pyramidal cells (PCs), has revealed their central role in generating network oscillations that underlie numerous cognitive processes[10–12]. Interestingly, ablation of NMDARs in PV INs results in altered network oscillations and emergence of behavioral deficits reminiscent of schizophrenic and autistic phenotypes[13–17]. These studies are part of a body of literature central to the tenet that PV IN dysfunction via perturbation of their NMDAR-signaling is causal in varying neurological

disorders. Intriguingly, this is puzzling since they express a relatively low, if not negligible, synaptic NMDAR component[18–22]. Furthermore, recent contradictory evidence[23] opens the possibility that NMDAR-signaling deficits at non-PV IN loci can also precipitate network dysfunction[24, 25]. Thus, a complete understanding of the cellular mechanisms underlying the development and synaptic integration of other IN subtypes should yield further insight into the etiology of such disorders.

Here, we focus on the neurogliaform family of INs[26, 27] that constitute the most abundant dendrite-targeting hippocampal INs[2] yet remains one of the most understudied. Hippocampal neurogliaform cells (NGFCs) can be subdivided with distinct cohorts originating from caudal and medial ganglionic eminences (CGE, MGE)[27, 28]. We employ a cre-lox approach to identify these respective NGFC populations and ablate their NMDARs. We provide a comprehensive examination of synaptic NMDAR expression and using an optogenetic strategy dissect for the first time the entorhinal and thalamic synaptic input onto these NGFCs. Interestingly, the activity of the long-range afferents results in the largest NMDAR-mediated synaptic component observed onto any hippocampal IN tested to date[18]. We demonstrate that NMDARs are critical for developmental programs involved in appropriate expression of short-term plasticity, AMPA receptor (AMPAR) function and dendrite patterning. Additionally, the dendritic location of a subpopulation of NGFCs enable them to receive inputs from CA3 via the Schaffer-

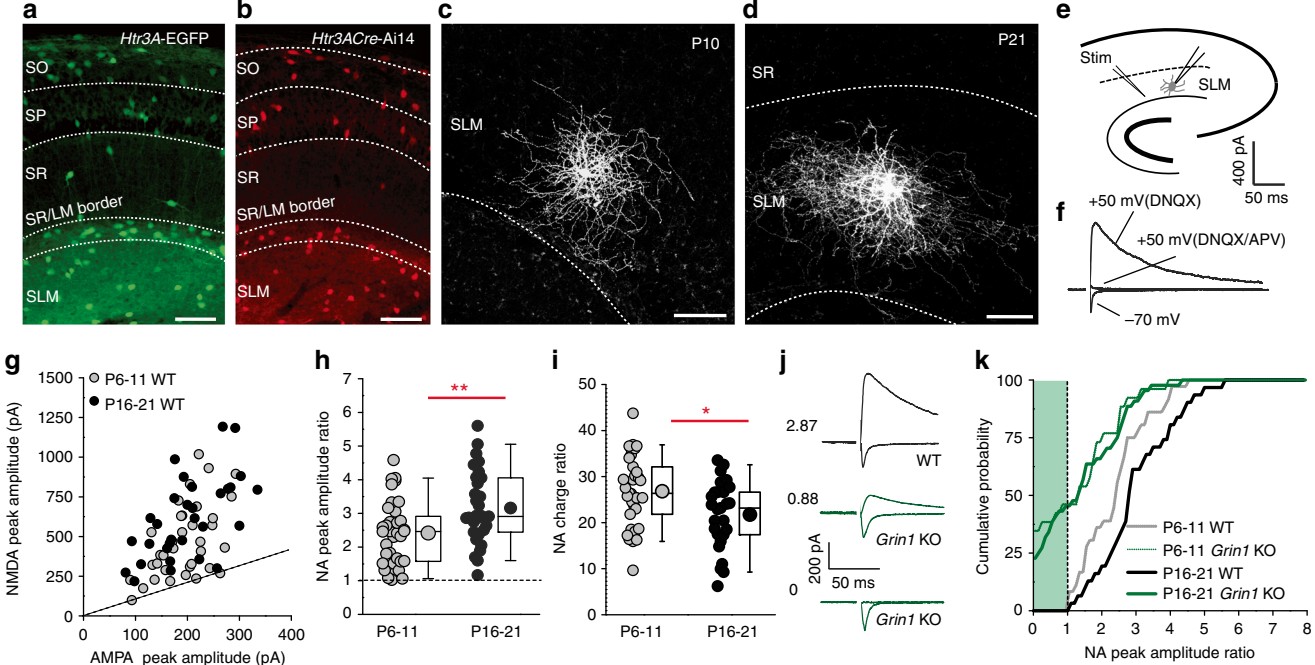

**Fig. 1** Postnatal developmental regulation and functional knockdown of NMDAR-mediated synaptic input onto CGE-derived NGFCs. **a, b** Confocal images of CA1 regions of hippocampi from *Htr3A*-EGFP and *Htr3A*-cre:Ai14 mice illustrating the lamina distribution of CGE-derived INs. *Scale bars*, 100 μm. **c, d** Confocal images of biocytin-filled SLM interneuron targeting using the reporter mice with NGFC morphology at P10 and P21 developmental time points. Scale bars, 50 μm. **e** *Schematic* depicting the recording configuration and stimulus (stim) employed to evoke glutamatergic synaptic input. **f** Representative example of a whole-cell voltage-clamp recording from the NGFC illustrated in **d** illustrating AMPAR and NMDAR-mediated EPSCs in response to SLM afferent stimulation. **g** *Scatterplot* of AMPAR- and NMDAR-mediated EPSC peak amplitudes in individual SLM NGFCs targeted using the *Htr3A*-cre:Ai14 mouse (referred subsequently as simply WT) at two developmental stages P6-11 (*gray circles*; n = 36) and P16-21 (*black circles*; n = 31). **h** NMDAR/AMPAR (NA) peak amplitude ratios in NGFCs at P6-11 (*gray circles*; n = 36) vs. P16-21 (*black circles*; n = 31). **i** NA charge ratio at P6-11 (*gray circles*; n = 36) vs. P16-21 (*black circles*; n = 31). **j** Single trace examples of NMDAR and AMPAR EPSCs in individual NGFCs in WT (*black traces*) and *Htr3A*-cre:Ai14:Grin1^flox/flox^ mouse lines (referred hereafter as *Grin1* KO; green traces). Values denote NA peak amplitude ratio in each individual cell. **k** Cumulative distribution of NA peak amplitude ratios in WT (*black lines*; n = 31 and 36 for P6-11 and P16-21, respectively) and *Grin1* KO (*green lines*; n = 26 and 46 for P6-11 and P16-21, respectively). *Dotted lines* in **g**, **h** and green shaded box in **k** denotes NA peak amplitude ratios of 1 or lower. Mann–Whitney *U*-tests were used for all comparisons (**p* < 0.05, ***p* < 0.01, ****p* < 0.001). CGE caudal ganglionic eminence, *SLM* stratum laconosum moleculare, *SO* stratum oriens, *SP* stratum pyramidale, *SR* stratum radiatum. All *n* values correspond to the number of cells recorded. Construction of *box-whisker plots* is detailed in methods

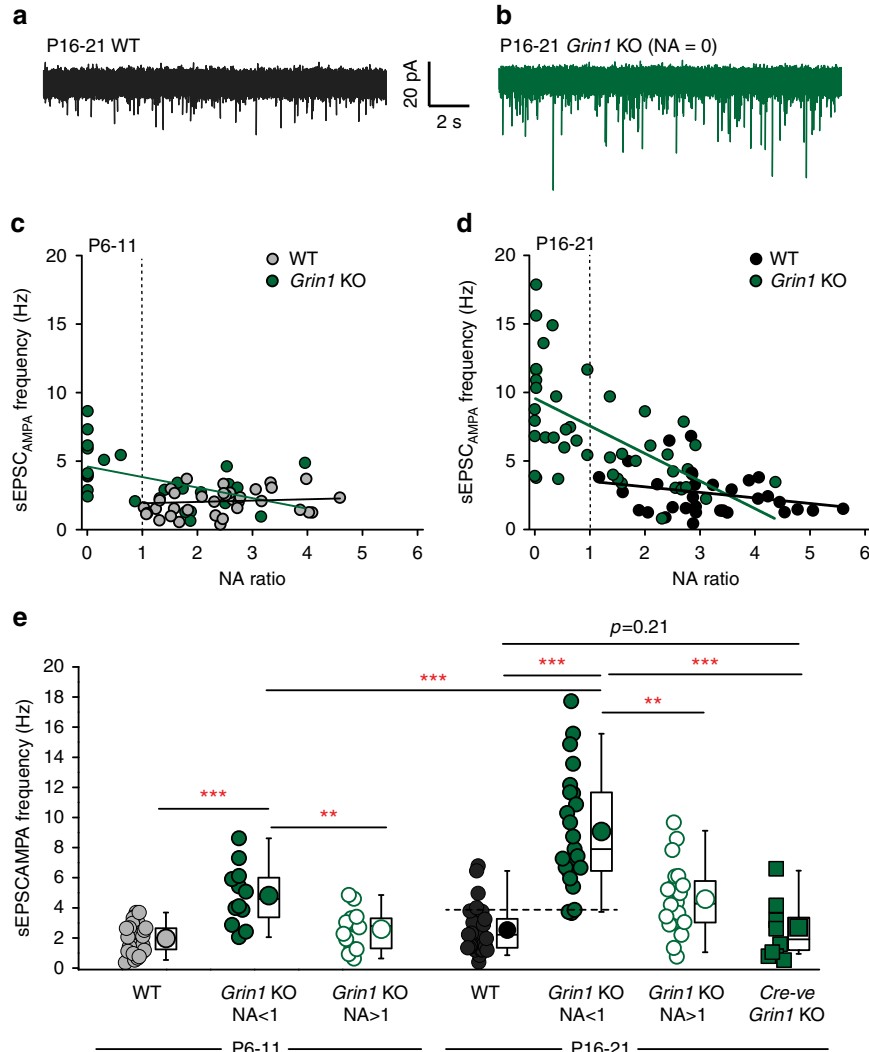

**Fig. 2** NMDAR-hypofunction precipitates an abnormal increase in sEPSC$_{AMPA}$ frequency in CGE-derived NGFCs. **a**, **b** Gap-free voltage-clamp traces illustrating sEPSC$_{AMPA}$ events recorded from a putative SLM NGFC in WT (*black trace*) and *Grin1* KO (*green trace*) mouse. **c**, **d** *Scatterplots* illustrating the relationship between NA ratio and sEPSC$_{AMPA}$ frequency in individual CGE NGFCs from WT (*gray and black circles; n* = 35, 30 for P6-11 and P16-21, respectively) and *Grin1* KO mice (*green circle; n* = 26, 43 for P6-11 and P16-21, respectively). **e** sEPSC$_{AMPA}$ frequency onto CGE NGFCs in WT (*gray and black circles; n* = 35, 30 for P6-11 and P16-21, respectively) and Grin1 KO mice (binned according to NA ratio; *filled green circles* for NA < 1 and *n* = 12, 23 at P6-11 and P16-21, respectively; *open green circles* for NA > 1 and *n* = 14, 20 at P6-11 and P16-21, respectively). *Green filled squares* denote sEPSC$_{AMPA}$ frequency measured in cre-negative cells in P16-21 Grin1 KO mice (*n* = 10). Mann–Whitney *U*-tests were used for all comparisons (***p* < 0.01, ****p* < 0.001). All *n* values correspond to the number of cells recorded. Construction of *box-whisker plots* is detailed in methods. *Dotted line* in **e** indicates sEPSC frequency > 4 for virtually all CGE NGFCs with NMDAR-hypofunction

collateral (SC) path[29–31]. Remarkably, NMDAR-ablation does not affect the functional properties of this input resulting in a shifted balance of the excitatory recruitment mediated by extra- vs. intra-hippocampal afferents that impinge on an individual NGFC. Taken together we demonstrate a critical, yet afferent specific, role for NMDARs in the appropriate integration of NGFCs into the hippocampal circuitry. These observations constitute previously undescribed synaptic pathophysiology involving NGFCs that likely contribute to circuit dysfunction potentially underlying the cognitive and behavioral deficits associated with neurodevelopmental disorders.

## Results

**Developmental expression and ablation of NMDARs in CGE NGFCs.** We initially focused on the major cohort of hippocampal NGFCs originating from the CGE and to this end employed the

*Htr3A*-EGFP and *Htr3A*-cre transgenic mouse lines with the latter crossed to a floxed tdTom mouse line (Ai14). These lines label CGE-derived INs from early developmental time points through to adulthood[32, 33]. Numerous EGFP-expressing and tdTom-expressing cells can be identified with similar distribution profile across the various hippocampal lamina (Fig. 1a, b). We targeted reported cells with small soma (~10 μm in diameter) in superficial regions of CA1 stratum laconosum-moleculare (SLM) at two developmental periods (postnatal day (p)6-11 and p16-21) for electrophysiological analyses. Post hoc biocytin reconstruction revealed that the majority of cells targeted (Methods section) possess morphology typical of NGFCs (i.e., very dense axonal arborization outspanning the relatively short dendrites; Fig. 1c, d). Their dendritic arbors were confined to the SLM indicating that all afferent input must be restricted to this region (Fig. 1c, d). Upon electrical stimulation of SLM afferents (Fig. 1e), we observed a remarkably large NMDAR/AMPAR (NA) peak

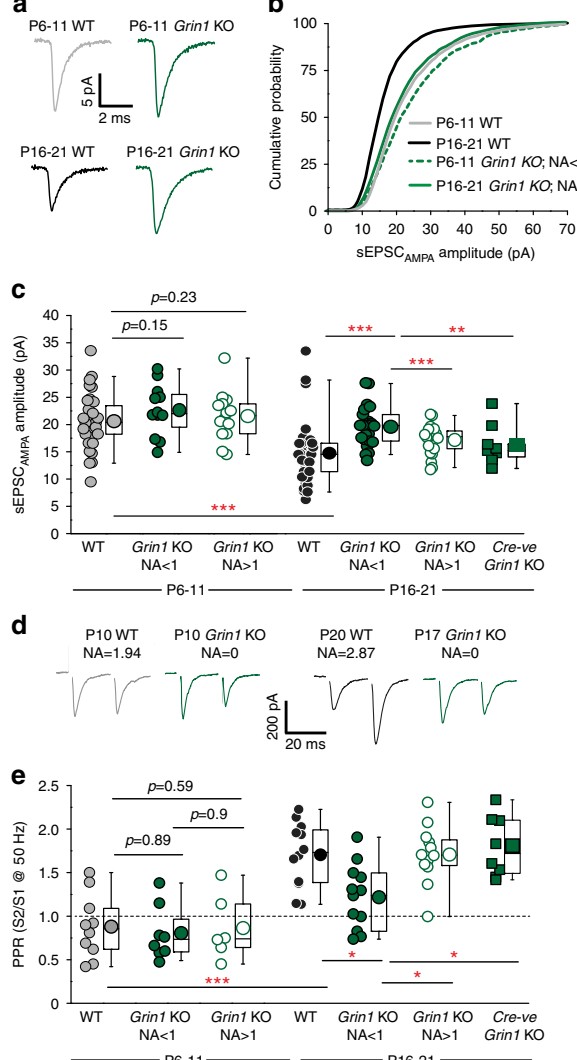

**Fig. 3** NMDAR-hypofunction retards normal postnatal development of sEPSC$_{AMPA}$ and SLM afferent evoked paired pulse ratio on CGE-derived NGFCs. **a** Ensemble averages of sEPSC$_{AMPA}$ events in WT (*gray* and *black traces*) and *Grin1* KO (*green traces*) CGE NGFCs at P6-11 and P16-21. **b** Cumulative distribution curves of sEPSC$_{AMPA}$ amplitude in WT (*gray* and *black lines; n* = 36, 31 for P6-11 and P16-21, respectively) and *Grin1* KO (*dotted* and *solid green lines; n* = 12, 22 for P6-11 and P16-21, respectively) mice. **c** Pooled data of mean sEPSC$_{AMPA}$ amplitudes measured in CGE NGFCs from WT (*gray* and *black circles; n* = 36, 31 for P6-11 and P16-21, respectively) and *Grin1* KO mice (binned according to NA ratio; *filled green circles* for NA < 1 and *n* = 12, 22 at P6-11 and P16-21, respectively; *open green circles* for NA > 1 and *n* = 13, 23 at P6-11 and P16-21, respectively). *Green filled squares* denote sEPSC$_{AMPA}$ amplitude measured in cre-negative cells in P16-21 *Grin1* KO mice (*n* = 10). (**d**) Single trace examples of evoked AMPAR-mediated EPSCs following paired pulse stimulation (50 Hz) of SLM afferent fibers in WT (*gray* and *black traces*) and *Grin1* KO mice (*green*) at p6-11 and p16-21. Corresponding ages and NA ratio in each NGFC is indicated above traces. **e** Paired pulse ratio (PPR) in WT (*gray* and *black circles; n* = 10, 12 for P6-11 and P16-21, respectively) and *Grin1* KO mice (binned according to NA ratio; *filled green circles* for NA < 1 and *n* = 8, 12 at P6-11 and P16-21, respectively; *open green circles* for NA > 1 and *n* = 6, 13 at P6-11 and P16-21, respectively). *Green filled squares* denote PPR measured in cre-negative cells in P16-21 Grin1 KO mice (*n* = 8). Mann–Whitney *U*-tests were used for all comparisons (\*\**p* < 0.01, \*\*\**p* < 0.001). All *n* values correspond to the number of cells recorded. Construction of *box-whisker plots* is detailed in methods

amplitude ratio (Fig. 1f–h) that increased during development. (Fig. 1g, h). These data demonstrate that hippocampal CGE-derived NGFCs possess the largest expression of functional synaptic NMDARs relative to AMPARs of all hippocampal INs tested to date (NA ratio ~3 and 10 times greater than that seen in previously characterized hippocampal CGE and MGE-derived INs, respectively[18]).

Interestingly, NMDAR-mediated EPSCs possess faster decay kinetics as development proceeds (Supplementary Fig. 1a, d) attributable to a decrease in NR2B expression as revealed by a lower sensitivity to the NR2B antagonist, ifenprodil (IF) (Supplementary Fig. 1b, c, e). Indeed, IF application significantly decreases the time constant of the decay at early postnatal stages, with minimal effect in juvenile mice (Supplementary Fig. 1f). Thus, CGE-derived NGFCs undergo postnatal development in which NR2B are replaced with NR2A subunits. Interestingly, although NA ratios are lower earlier in development (Fig. 1h), the slower NR2B-containing NMDAR kinetics at this stage endows NGFCs with significantly larger NA charge ratios (Fig. 1i). To determine the physiological role of NMDARs with regards NGFC recruitment and action potential (AP) output we monitored SLM-evoked EPSPs in NGFCs with the characteristic late-spiking phenotype (Supplementary Fig. 2a). EPSPs summated during trains of SLM stimulation that was attenuated after pharmacological block of NMDARs with the NMDAR EPSP influence on summation greatest at higher stimulation frequencies and during the latter parts of the train (Supplementary Fig. 2a, b). Finally, NMDARs significantly alter the timing of NGFC APs by influencing their latency and jitter[18, 34] (Supplementary Fig. 2c–e). Thus, the SLM afferent input onto NGFCs is characterized by a heavy bias towards NMDAR signaling and this large contribution during postnatal development through juvenile stages is involved in shaping the dynamics of synaptic input/output of NGFCs.

We next investigated the potential role of NMDARs on the functional development of NGFCs. We ablated NMDARs in CGE NGFCs by crossing the *Htr3A*-cre:Ai14 mouse line to the *Grin1* floxed KO mouse. We determined the penetrance of functional synaptic NMDAR KO by analyzing NA ratios in TdTom positive (i.e., cre-expressing) NGFCs in this *Htr3A*-cre:Ai14:*Grin1*$^{flox/flox}$ mouse line (referred hereafter as *Grin1* KO). Upon SLM stimulation, NA ratios in NGFCs of *Grin1* KO were lower than in the WT mice at early postnatal and juvenile stages (Fig. 1j, k). The NA ratio distribution was essentially similar at P6-11 and P16-21 in the *Grin1* KO mice (Fig. 1k). Approximately a quarter of NGFCs possess no NMDAR-mediated EPSC (i.e., NA ratio of 0) demonstrating complete ablation of functional NMDARs (Fig. 1k). Another quarter present with NA ratios below 1, a value never observed in NGFCs recorded from WT mice, pointing towards an incomplete but nevertheless functional knockdown (Fig. 1k). Finally, a number NGFCs possess NA ratios > 1 demonstrating that not all undergo marked NMDAR ablation using the cre-lox strategy outlined (Fig. 1k). Thus, the genetic strategy employed results in NMDAR-hypofunction throughout postnatal development but only in a subpopulation of cre-expressing CGE-derived NGFCs.

**Progressive and abnormal increase in AMPAR mediated input.** We next examined the effects of NMDAR hypofunction on the synaptic integration of NGFCs into the circuitry by analyzing the frequency of spontaneous AMPAR-mediated EPSCs (sEPSC$_{AMPA}$). In *Grin1* KO mice a clear relationship was apparent between sEPSC$_{AMPA}$ frequency and NA ratio with the highest sEPSC$_{AMPA}$ frequency values observed in NGFCs with total NMDAR ablation (NA = 0; Fig. 2a–c). Binning the data according to NA ratio showed that at P6-11, complete abolition

or functional knockdown of NMDARs (i.e., NA < 1) result in a significant increase of sEPSC$_{AMPA}$ frequency that exacerbates as postnatal development progresses (Fig. 2d). One possible reason for this abnormality is that NMDAR ablation in *Htr3A*-cre-expressing INs precipitates secondary changes in global network activity. Of course, our genetic strategy also impacts a variety of CGE-derived hippocampal IN subtypes that could precipitate network wide alterations. We therefore took advantage of the fact, as explained above, that a subset of cre-expressing NGFCs in the *Grin1* KO mice possess NA > 1 indicative of incomplete penetrance of NMDAR ablation. In these NGFCs, sEPSC$_{AMPA}$ frequencies were significantly lower than that seen in NGFCs with NA < 1 (Fig. 2e). In addition, further analysis of Cre-expression was undertaken by crossing the *Htr3A*-cre:Ai14 and *Htr3A*-EGFP mouse lines. In the SLM although virtually all TdTom positive cells expressed EGFP + only half of the EGFP + cells express TdTom (Supplementary Fig. 3). Thus ~50% of CGE-derived INs (Supplementary Fig. 3) including NGFCs do not express cre-activity and are unable to undergo conditional deletion of *Grin1*. We therefore targeted TdTom-negative (i.e., Cre-negative) putative NGFCs in P16-21 *Grin1* KO *mice* (labeled hereafter as Cre-ve *Grin1* KO). Interestingly, sEPSC$_{AMPA}$ frequency was similar to WT and significantly lower than that seen in NGFCs that have undergone NMDAR-hypofunction (Fig. 2e). Together the ability to perform such in-slice controls in KO mice, clearly demonstrate that NMDARs in hippocampal NGFCs serve to negatively regulate AMPAR-mediated input during postnatal development in a cell autonomous manner, similar to that previously observed in hippocampal PCs[35, 36].

**Unaltered intrinsic properties of NGFCs after NMDAR ablation**. TdTom positive NGFCs with late-spiking phenotype could be identified in P16-21 *Grin1* KO mice (Supplementary Fig. 4a, b) demonstrating that NMDAR hypofunction does not impact expression of intrinsic conductances responsible for this phenomenon. The sEPSC$_{AMPA}$ frequency was measured in these late-spiking NGFCs and in agreement with our previous data were found to be significantly higher than that seen in WT NGFCs (Supplementary Fig. 4c). Since NA ratios in these current-clamp experiments could not be assessed, sEPSC$_{AMPA}$ frequency was used as a proxy for NMDA hypofunction due to our previous observed correlation (Fig. 2c, d). We previously showed that the majority of NGFCs with NA < 1 displayed a sEPSC$_{AMPA}$ frequency greater than 4 Hz, a value not evident in P16-21 WT NGFCs (Fig. 2e; see *dotted line*). We therefore analyzed membrane properties in late-spiking NGFCs that demonstrated sEPSC$_{AMPA}$ frequency greater than this value in *Grin1* KO mice on the assumption that they underwent significant NMDAR-hypofunction. When compared to NGFCs in WT mice, no significant differences in basic membrane and firing parameters were apparent (Supplementary Table 1). These data illustrate that the expression of the numerous ion channels responsible for these physiological properties develops normally in the absence of NMDAR function demonstrating that NMDAR-signaling is not promiscuously involved in all aspects of their cellular development.

**Abnormal development of synaptic function upon NMDAR ablation**. To further interrogate the impact of NMDAR hypofunction on the synaptic integration of CGE NGFCs we analyzed the amplitude of sEPSC$_{AMPA}$ during development. In WT mice a leftward shift of the cumulative distribution and corresponding reduction in the mean sEPSC$_{AMPA}$ amplitude onto NGFCs is observed between P6-11 and P16-21 (Fig. 3a–c). In P6-11 *Grin1* KO mice, NMDAR hypofunction does not significantly

alter sEPSC$_{AMPA}$ amplitude (Fig. 3a–c). However, at P16-21, sEPSC$_{AMPA}$ amplitudes in *Grin1* KO mice are significantly increased when compared to their age-matched controls with cumulative distribution and mean value reminiscent of that seen in WT P6-11 mice (Fig. 3a–c). Thus, in the absence of NMDAR signaling the normal developmental trajectory of this post-synaptic parameter is impaired. This deficit is cell autonomous in nature since a significant difference in sEPSC$_{AMPA}$ amplitudes, to values closer to P16-21 WT, is noted between NGFCs in P16-21 *Grin1* KO displaying NMDAR-hypofunction (NA < 1) and those that possess NA > 1 or are Cre-negative in the same mouse (Fig. 3c).

The paired pulse ratio (PPR), a measure of presynaptic function employed to assess probability of release (Pr), of SLM inputs onto NGFCs in WT mice significantly changes from depressing to facilitating between P6-11 and P16-21 (Fig. 3d, e). Thus, a conversion from high to low Pr synapses occurs during development in a similar manner to that described at SLM inputs onto CA1 PCs[37]. In P6-11 *Grin1* KO mice the PPR at NGFCs with NMDAR hypofunction (i.e., NA < 1) was unchanged compared to WT mice or in NGFCs with NA > 1 from *Grin1* KO mice (Fig. 3e). However, in P16-21 *Grin1* KO mice, NGFCs with NA < 1 demonstrated a PPR significantly lower than in age-matched WT with values trending towards those seen at P6-11 (Fig. 3d, e). Therefore, NMDAR function is critical in ensuring the normal development trajectory of Pr at SLM inputs to NGFCs. Furthermore, at the latter developmental stage tested, the PPR of SLM inputs onto NGFCs in WT mice was not significantly different to those found in Cre-positive NGFCs from *Grin1* KO mice that displayed NA > 1 nor NGFCs in this mouse that were Cre -ve (Fig. 3e). Thus, as with the observed alterations in sEPSC$_{AMPA}$ frequency and amplitude, the changes in PPR precipitated by NMDAR-hypofunction also demonstrates cell autonomy.

Taken together our analyses over the first 3 postnatal weeks clearly indicate NMDAR-expression is critical for specific aspects of normal synaptic development at both pre- and post-synaptic loci.

**Synaptic underpinnings of the exuberance in sEPSC frequency**. Since Pr can influence sEPSC frequency we examined whether maintenance of a high Pr following NMDAR hypofunction underlies the increased sEPSC frequency observed above. Interestingly, unlike in P6-11 *Grin1* KO mice, at P16-21 a clear relationship exists between PPR and sEPSC$_{AMPA}$ frequency in individual NGFCs linking the observed aberrant increase in sEPSC$_{AMPA}$ frequency with changes in presynaptic function (Supplementary Fig. 5). However, a number of additional observations based on correlations of sEPSC$_{AMPA}$ frequency and PPR suggest that NMDAR-hypofunction mediated alterations in Pr is not the sole contributor of the augmented sEPSC$_{AMPA}$ frequency. First, in WT mice although a large decrease in Pr is noted between P6-11 and P16-21(Fig. 3d, e), sEPSC$_{AMPA}$ frequency does not alter (Fig. 2c–e) suggesting that during normal development other synaptic mechanisms are at play that offset the influence of Pr on sEPSC$_{AMPA}$ frequency (Supplementary Fig. 5c; *gray arrow*). Second, NGFCs at P6-11 with NMDAR hypofunction exhibit an enhanced sEPSC$_{AMPA}$ frequency in the absence of changes in PPR relative to WT (Supplementary Fig. 5c; *solid black arrow*) illustrating that at this developmental age this increase is independent of alterations in Pr. Finally, in NGFCs with NMDAR-hypofunction at P16-21 although Pr is markedly higher than in WT mice it is still considerably lower than that seen at P6-11, an age where the increase in sEPSC$_{AMPA}$ frequency following NMDAR hypofunction is not nearly as robust

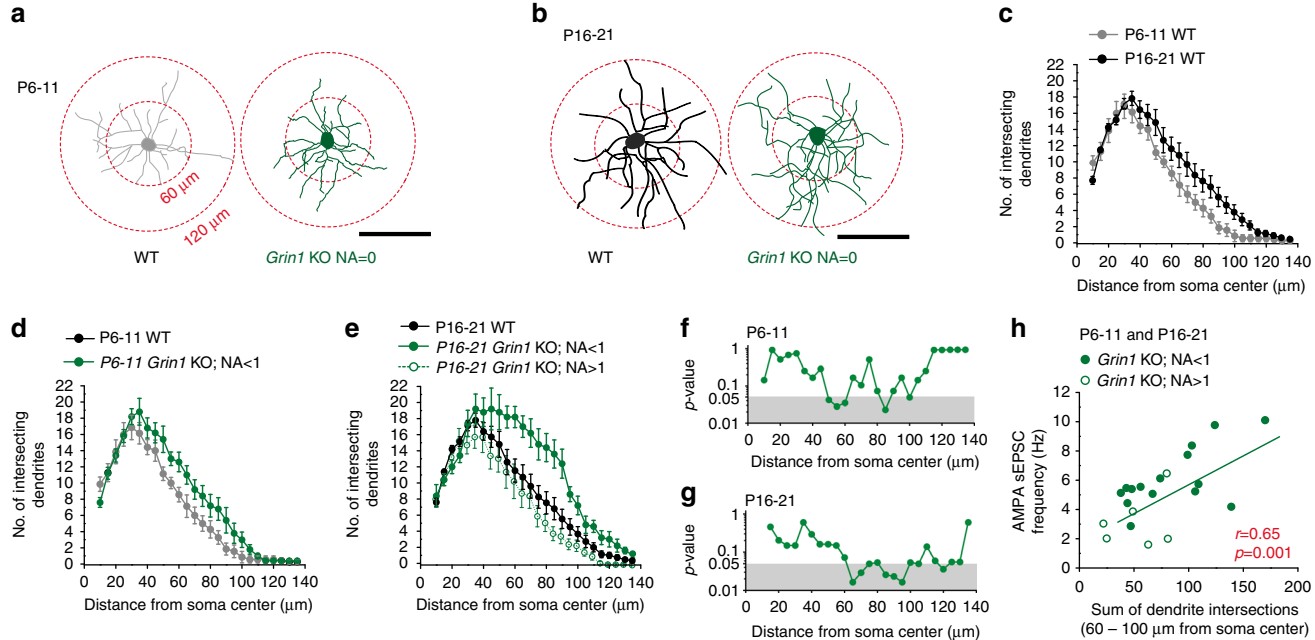

**Fig. 4** NMDAR-hypofunction induces an increased dendrite complexity of CGE-derived NGFCs during postnatal development. **a, b** Single cell examples of the dendritic arborization of WT (*gray* and *black* cells; P6-11 and P16-21, respectively) and *Grin1* KO (*green cells*) CGE NGFCs. Examples of concentric circles employed for subsequent Sholl analyses (*red circles*). Scale bars, 100 μm. **c** Sholl analyses plots of total number of dendrites intersecting each concentric circle (0–135 μm from center of soma every 5 μm) in P6-11 (*gray data*; n = 7) and P16-21 (*black data*; n = 15) WT CGE NGFCs. **d** Sholl analyses plots comparing morphology of P6-11 WT (*gray data*; n = 7) and Grin1 KO (*green data*; NA < 1; n = 6) CGE NGFCS. Note, P6-11 WT data (*gray*) in this panel is re-plotted from **c**. **e** Sholl analyses plots comparing morphology of P16-21 WT (*black data*; n = 15) and *Grin1* KO CGE NGFCS (*green data*; filled vs. open circles denote data from NGFCs with NA < 1 and NA > 1, respectively; n = 11 and 6). Note, P16-21 WT data (*black*) in this panel is re-plotted from **c**. **f, g** Statistical comparison of the number of intersecting CGE NGFC dendrites at each concentric distance away from cell body between WT and *Grin1* KO mice (*p*-values are derived from Mann–Whitney *U*-test; *Gray shaded* area denotes *p*-values < 0.05). **h** Correlation of the sum of dendrite intersection (60–100 μm from the soma center) and sEPSC$_{AMPA}$ frequency in individual NGFCs. *r* and *p*-values as indicated from straight-line fit statistical analysis demonstrates a significant positive linear correlation between dendrite intersections and sEPSC$_{AMPA}$ frequency. All *n* values correspond to the number of cells recorded. Error bars denote SEM

(Supplementary Fig. 5c; *dotted black and green arrows*). In summary, although a correlation between the sEPSC$_{AMPA}$ frequency and PPR exists at P16-21 (Supplementary Fig. 5b), this cannot fully underlie the abnormal AMPAR-mediated synaptic input precipitated by NMDAR-hypofunction.

The total number of synapses expressed also contributes to sEPSC$_{AMPA}$ frequency. However, unlike PCs the vast majority of INs, including NGFCs, do not possess dendritic spines and thus such anatomical substrates cannot be employed to delineate their putative functional synapses. We therefore analyzed the morphology of NGFC dendritic arbors as a measure of potential surface area for synaptic input. Sholl analyses indicate a small increase in dendritic complexity during normal postnatal development (Fig. 4a–c). Interestingly, one can postulate that these changes in dendrite arborization, indicative of an increase in putative functional synapses, counteracts the decreased Pr noted across the same developmental epoch (Fig. 3d, e) resulting in the relatively stable sEPSC$_{AMPA}$ frequency observed (Fig. 2e; Supplementary Fig. 5c; *gray arrow*). In contrast, NMDAR hypofunction precipitates a significant increase in dendritic complexity when compared to WT that commences at P6-11 escalating as development proceeds (Fig. 4d–f). As with the sEPSC$_{AMPA}$ frequency and PPR alterations the morphological exuberance occurred in a cell autonomous manner since NGFCs with NA > 1 in *Grin1* KO mice displayed similar dendritic arbors to their WT counterparts (Fig. 4e). Interestingly at P16-21, similar to the relationship to PPR (Supplementary Fig. 5b), a significant correlation between dendrite complexity and sEPSC$_{AMPA}$ frequency in individual NGFCs following NMDAR ablation is

apparent (Fig. 4h). Thus, although a direct assessment of synapse number would be preferable, this correlation indicates dendrite complexity provides a suitable proxy for number of functional synapses. With this assumption in mind, we propose that during the earliest time point investigated (P6-11) the observed NMDAR-hypofunction mediated increase in sEPSC$_{AMPA}$ frequency (Fig. 2c, e) occurring in the absence of changes in PPR (i.e., Pr) (Fig. 3d, e; Supplementary Fig. 5a) may be a consequence of an increase in functional synapse number. As development proceeds under continued *Grin1* ablation, the increase in functional synapses is more pronounced than at the earlier developmental epoch (Fig. 4e, g). At this developmental stage, this increase in combination with the maintenance of high Pr at excitatory inputs onto NGFCs (Fig. 3d, e) results in a larger, more robust abnormal increase in sEPSC$_{AMPA}$ frequency (Fig. 2d, e).

Together our analyses demonstrate abnormalities at both pre- and postsynaptic loci emerge at different postnatal stages to ultimately combine producing the aberrant increase in synaptic input onto NGFCs following ablation of NMDAR function.

**Similar abnormalities observed in MGE-derived NGFCs.** An additional NGFC population derived from MGE progenitors is present within the hippocampus[28]. Since a number of developmental programs and functional properties[18, 28, 38–40] of INs relate to their embryonic origin we asked whether NMDARs play similar roles in the synaptic integration of these MGE-derived NGFCs. We utilized the Nkx2.1cre mouse line that fate-maps

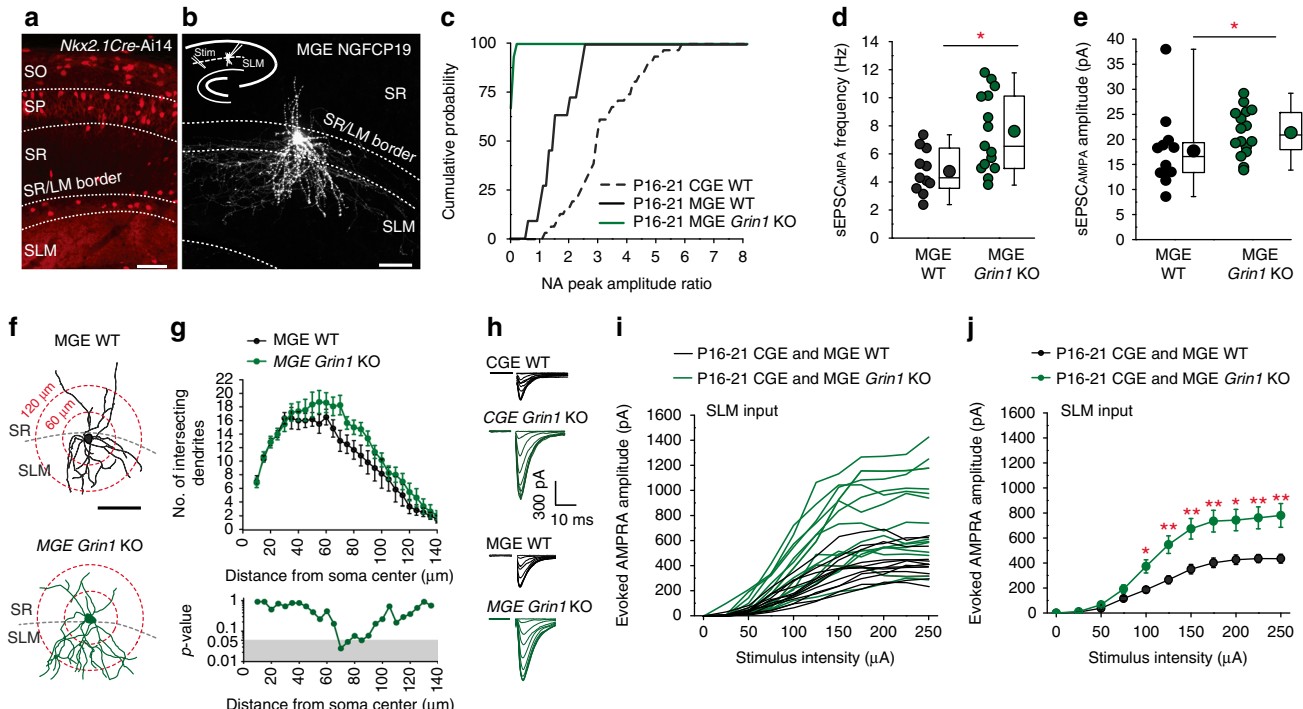

**Fig. 5** NMDA-hypofunction precipitates similar morpho-physiological abnormalities in MGE NGFCs resulting in increased synaptic strength of SLM excitatory drive regardless of embryonic origin. **a** Confocal image of Nkx2.1Cre:Ai14 mouse illustrating the lamina distribution of MGE-derived interneurons. *Scale bar*, 100 μm. **b** Confocal image of a biocytin-filled TdTom-positive NGFC illustrating the polarized dendrites in SLM and SR. *Scale bar*, 50 μm. *Inset* depicting recording and stimulus (stim) electrode arrangement. **c** Cumulative distribution of NA peak amplitude ratios in P16-21 MGE WT (*black line*; n = 12) and MGE *Grin*1 KO (*green line*; n = 16) mice. *Black dotted line* denotes NA ratio distribution in CGE WT NGFCs at P16-21 and is re-plotted from Fig. 1k for comparative purposes. **d** sEPSC$_{AMPA}$ frequency in MGE NGFCs in P16-21 WT (*black circles*; n = 11) and *Grin*1 KO mice (*green circles*; n = 16). **e** sEPSC$_{AMPA}$ amplitude in MGE NGFCs in P16-21 WT (*black circles*; n = 11) and *Grin*1 KO mice (*green circles*; n = 16). **f** Dendritic arborization of MGE NGFCs from WT (*black cell; top panel*) and *Grin*1 KO (*green cell; bottom panel*) mice. Examples of concentric circles employed for Sholl analyses (*red circles*). *Scale bar*, 100 μm. **g** Sholl analyses plots of NGFC dendrites from WT (*black data*; n = 6) and *Grin*1 KO mice (*green data*; n = 12). *Bottom panel* show statistical comparison with *shaded gray areas* denoting p < 0.05. **h** Single overlaid traces of SLM afferent evoked AMPAR-mediated EPSCs in CGE and MGE NGFCS from P16-21 WT (*black traces*) and *Grin*1 KO (*green traces*) mice at varying stimulus strengths (0–250 μA in 25 μA steps). **i, j** Individual and pooled relationship of SLM-evoked AMPAR-mediated EPSC amplitude vs. stimulus strength. Data of P16-21 CGE and MGE NGFCs from WT (*black data*; n = 7 and 6, respectively) and *Grin*1 KO (*green data*; n = 8 and 7, respectively) mice are combined. SO stratum oriens, SLM stratum laconosum molecular, SP stratum pyramidale, SR stratum radiatum, Mann–Whitney U-tests were used for all comparisons (*p < 0.05, **p < 0.01). Construction of *box-whisker plots* is detailed in methods. All n values correspond to the number of cells recorded. *Error bars* denote SEM

MGE progenitors[41] that give rise to hippocampal INs including MGE NGFCs[28, 30]. Thus, similar to the genetic strategy targeting CGE-derived NGFCs, NMDAR-function can be ablated in their MGE counterparts from early embryonic developmental stages[41]. TdTom positive hippocampal INs in the Nkx2.1cre:Ai14 mice are biased towards the deeper layers and are rarely seen in the SLM region (Fig. 5a). Restricting our analyses to the P16-21 developmental time point, we targeted TdTom positive cells with small soma found at the SLM/SR border region and NGFC identity confirmed with *post hoc* biocytin reconstruction (Fig. 5b)[28, 29, 30].

We determined the NA ratio of SLM-evoked EPSCs in MGE NGFCs. The distribution of NA ratios was leftward shifted (Fig. 5c) and the mean NA ratio was lower when compared to their CGE counterparts (mean NA ratio = 1.6 ± 0.2 and 3.2 ± 0.2 for MGE vs. CGE NGFCs, respectively; n = 12 and 31, respectively; p value = 0.00001; Mann–Whitney U-test). However, the NA ratio in MGE NGFCs is markedly higher than that previously described for other MGE-derived hippocampal IN subtypes[14, 18, 42] demonstrating that SLM afferent input onto NGFCs of both embryonic origins are characterized by large synaptic NMDAR component. We next asked whether NMDAR ablation in MGE NGFCs also altered their synaptic integration

and to this end crossed the *Nkx2.1*cre:Ai14 mice with the floxed *Grin*1 KO line (MGE *Grin*1 KO). We confirmed that NMDAR function was abolished as evidenced by the distribution of NA ratio in MGE NGFCS in these mice (Fig. 5c; mean NA ratio in MGE *Grin*1 KO mice = 0.06 ± 0.02; n = 12). As previously noted in CGE NGFCs (Fig. 2d, e; Fig. 3a–c), NMDAR ablation in MGE NGFCs resulted in statistically significant higher AMPAR-mediated sEPSC$_{AMPA}$ frequency and mean amplitudes at P16-21 (Fig. 5d, e). Additionally, the total dendritic arborization of MGE NGFCs reveal a modest (when compared to that seen of P16-21 CGE NGFCs; Fig. 4b, e, g) yet significant increase in complexity at certain regions of the dendritic field (Fig. 5f, g). Thus, although an extensive analysis of the effects of NMDAR ablation on the synaptic integration across development was not undertaken, our data demonstrate that similar abnormalities are ultimately precipitated regardless of the embryonic origin of NGFCs.

Evoked EPSC amplitude provides a measure of the overall synaptic strength onto NGFCs and a significant increase in this parameter was observed in NGFCs displaying total NMDAR ablation (NA = 0) when compared to WT mice across a range of stimulus intensities including those (>175 μA) that elicit a maximal evoked AMPAR-mediated EPSC amplitude (since essentially the same effects were noted, data were pooled from

both CGE and MGE NGFCs; Fig. 5h–j). Thus, upon NMDAR ablation common alterations in sEPSC frequency and dendritic architecture suggestive of more functional synapses coupled with larger sEPSC$_{AMPA}$ amplitude indicating increased AMPARs per synapse clearly result in an abnormal strengthening of AMPAR-mediated excitatory drive of NGFCs elicited by SLM afferents impinging on both cohorts of NGFCs.

**Afferent specific increase in excitatory synaptic strength.** The anatomical position of NGFC dendrites in the SLM of the hippocampus endows them with the ability to receive extra-hippocampal temporoammonic pathway (TA) input arising from layer III medial entorhinal cortex[26] (mEC). In addition, afferents from ventral midline thalamic nuclei, including nucleus reuniens (NRe) provide another major extra-hippocampal input to this region[43] and has been shown to effectively recruit inhibition[44]. However, to date these respective afferent inputs onto hippocampal INs, including NGFCs, have not been investigated in isolation. Our experiments thus far probing sEPSC$_{AMPA}$ and evoked AMPAR-mediated EPSCs evoked by electrical SLM stimulation does not permit differentiation between these distinct afferent inputs. To dissect these distinct excitatory inputs, we delivered channel rhodopsin (ChR2) by viral injection into regions of mEC or NRe (Methods section; Supplementary Fig. 6). After a minimum of 2 weeks post injection, infection of afferent fibers coursing through the SLM were clearly apparent (Fig. 6a–d). Note that in addition to the TA path, the medial perforant path in the molecular layer of the dentate gyrus is also infected after viral injection in mEC (Fig. 6b, d) as previously demonstrated[45]. EPSCs in SLM NGFCs were evoked following light activation of either mEC or NRe afferents and NA ratios onto both CGE and MGE NGFCs (Fig. 6e) assayed. At the former subgroup of NGFCs, we demonstrate that both TA and thalamic inputs both contribute to the large NA ratio (Fig. 6e, f) previously noted upon indiscriminate electrical stimulation of SLM afferents (Fig. 1f–h). In agreement with our electrical stimulation data (Fig. 5c) both inputs elicit an appreciable yet smaller NA ratio on to the MGE cohort of NGFCs when compared to CGE NGFCs (Fig. 6e, f). Thus, taken together we illustrate that although eliciting differing NA ratios dependent on embryonic origin of NGFCs, the distinct extra-hippocampal excitatory inputs produce a similar NA ratio onto a given NGFC.

We next asked whether the NMDAR-hypofunction mediated increase in synaptic strength revealed following electrical evoked SLM afferent stimulation (Fig. 5h–j) is attributable to changes at one or both of these specific inputs. Note that these experiments were performed at latter developmental ages (i.e., p35–p57) and we confirmed that the increase in sEPSC$_{AMPA}$ frequency persisted following NMDAR-hypofunction (sEPSC$_{AMPA}$ frequency in WT and Grin1 KO mice = $1.6 \pm 0.14$ and $7.3 \pm 0.9$, respectively; $n = 10$ and 15, respectively) into adulthood. We assessed the maximal EPSC amplitude evoked by light resulting in mEC or NRe afferent activation (Methods section). Since essentially similar results were obtained for both CGE and MGE NGFCs the data were pooled. We observed a significant doubling of maximal AMPAR-mediated EPSC amplitude of both mEC and NRe inputs onto NGFCs lacking functional NMDARs (Fig. 6h, i). Therefore, our optogenetic strategy demonstrates that the NMDAR-hypofunction mediated increase in synaptic strength onto NGFCs is ubiquitously expressed at both the cortical and thalamic excitatory inputs.

Finally, the anatomical profile of a subpopulation of hippocampal NGFCs residing on the SR/SLM border clearly reveals not only a dendritic arbor in SLM but also one pervading into the SR region[29–31] (Fig. 5b, Fig. 6c, d). We therefore asked whether the

morpho-physiological abnormalities at SLM dendrites/inputs observed to date in the study were mirrored in the SR dendrite compartment. First, we reanalyzed our Sholl data from NGFCs in WT and Grin1 KO mice that possessed dendrites in both hippocampal lamina mice. A striking selectivity of NMDAR-hypofunction mediated morphological changes were revealed (Fig. 7a–d). Specifically, there was a significant increase of SLM dendrite complexity, while the SR dendrites demonstrated a slight but non-significant trend towards a less complex arbor (Fig. 7a–d). Remarkably, in NGFCs with NMDAR ablation, the sEPSC$_{AMPA}$ frequency only statistically correlated with the complexity of SLM dendrites (Supplementary Fig. 7). Furthermore, by analyzing the relative changes in dendrite architecture within each individual NGFC in WT vs. Grin1 KO mice a significant increase in the ratio of SLM/SR dendrite complexity was observed (Fig. 7e).

The segregation of dendrites coupled with distinct placement of stimulating electrodes allowed for activation of synaptic inputs impinging either on SLM or SR portions of the dendritic arbor in the same NGFC (Fig. 7f). SLM stimulation resulted in paired pulse facilitation (Fig. 7f, g). Stimulation of SR afferents also resulted in paired pulse facilitation but the calculated PPR was significantly lower to that seen at SLM synapses (Fig. 7f, g) demonstrating a divergence of Pr that is dependent on afferent input. Remarkably, following NMDAR ablation PPR was only significantly altered at the SLM input but remained unchanged at the SR input (Fig. 7f, g). We previously showed that synaptic input of the extra-hippocampal inputs (i.e., entorhinal and thalamic) impinging on SLM dendrites of NGFCs is significantly strengthened following NMDAR ablation (Fig. 5h–j; Fig. 6g–i). In a subset of these NGFCs we also measured the SR afferent mediated maximal evoked AMPAR-mediated EPSC and demonstrate unchanged/slightly lowered synaptic strength at this distinct afferent input (Fig. 7h). Calculating the ratio of the evoked response across differing stimuli intensities upon SR and SLM afferent stimulation in each individual NGFC reveal that under WT conditions the SR afferents provide the dominant input that significantly shifts in favor of the SLM afferents upon NMDAR ablation (Fig. 7i). Thus, these data together demonstrate that NMDAR-hypofunction precipitates selective changes in the synaptic integration of an individual NGFC into the hippocampal circuitry dependent on whether the excitatory afferents arise from extra vs. intra-hippocampal sources.

**Discussion**
Here, we demonstrate that synaptic input onto hippocampal NGFCs is characterized by the largest synaptic NMDAR-mediated component described on any hippocampal IN to date. Importantly, NMDAR-signaling is required for normal development and expression of various synaptic properties of NGFCs. How do these current observations compare to previous studies in which NMDAR function has been genetically manipulated in other neuronal subtypes in various brain regions? Perturbations of NMDAR expression in hippocampal PCs during development result in an increase of AMPAR-mediated signaling[35, 36, 46]. However, these changes occur in the absence of alterations in dendritic morphology, spine density[35] and presynaptic release probability[35, 46]. Thus, although NMDARs play similar roles in hippocampal NGFCs and PCs, serving as a brake for AMPAR-signaling during development, the underlying mechanisms considerably diverge. To our knowledge only two other studies to date have examined the impact of genetic NMDAR perturbation during early postnatal development on their integration and physiology[47, 48]. In the hippocampus, INs located in SO, likely corresponding to a population of SOM and/or PV-expressing

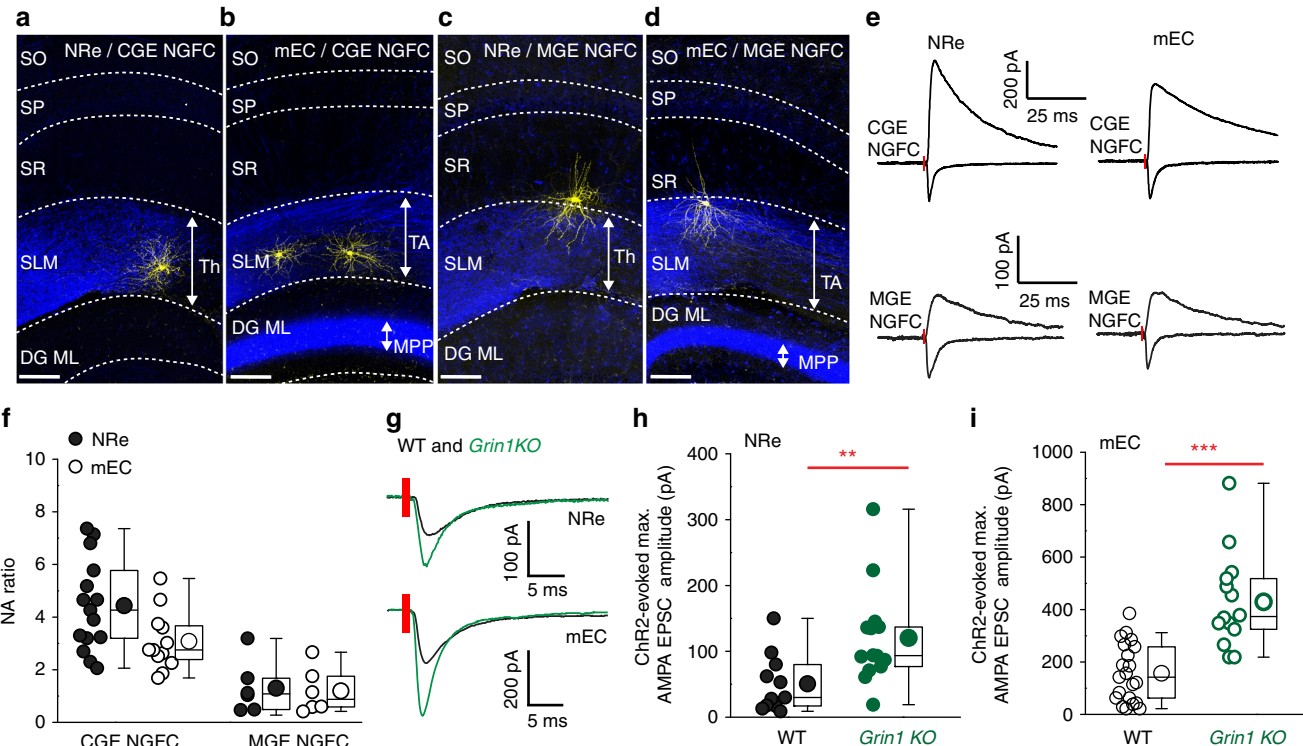

**Fig. 6** Optogenetic dissection reveals NMDAR hypofunction results in strengthening of both thalamic and cortical inputs onto NGFCs. **a–d** Confocal images of biocytin recovered CGE and MGE NGFCs in the hippocampus in WT mice injected with viral channelrhodpsin into the thalamic nucleus reuniens (NRe) or medial entorhinal cortex (mEC; Methods section for details). Note: images are pseduocolored for enhanced contrast. *Scale bars*, 100 μm. **e** Single trace examples of light-evoked AMPAR and NMDAR EPSCs in CGE and MGE NGFCs elicited by NRe and mEC afferent fibers. **f** NA peak amplitude ratios in CGE and MGE NGFCs resulting following light activation of thalamic (*filled circles*) or cortical afferent input (*open circles*; $n = 15$ and 6 for NRe inputs to CGE and MGE NGFCs, respectively; $n = 12$ and 6 for mEC inputs to CGE and MGE NGFCs, respectively). **g** Single trace examples of maximal channel rhodopsin evoked AMPAR-mediated EPSCs on NGFCs in WT (*black traces*) and *Grin1* KO mice (*green traces*) in CGE NGFCs. **h, i** Maximal channel rhodopsin evoked AMPAR-mediated EPSCs mediated by extra-hippocampal afferent input on NGFCs in WT (*black circles*; $n = 11$ and 21 for NRe vs. mEC mediated responses, respectively) and *Grin1* KO (*green circles*; $n = 14$ and 14 for NRe vs. mEC mediated responses, respectively) mice. *Red bars* on single example traces in **e** and **g** illustrate time of light stimulus. *DG ML* dentate gyrus molecular layer; *MPP* fiber tract of medial perforant path originating from mEC; *SLM* stratum laconosum moleculare; *SO* stratum oriens; *SP* stratum pyramidale; *SR* stratum radiatum; *TA* fiber tract of temporammonnic path originating from mEC; *Th* fiber tract originating from thalamic nuclei. Mann–Whitney *U*-tests were used for all comparisons (**$p < 0.01$, ***$p < 0.001$). All *n* values correspond to the number of cells recorded. Construction of *box-whisker* plots is detailed in Methods section

subtypes, *Grin2B*-ablation resulted in fewer AMPAR-containing synapses and a reduced AMPAR EPSC frequency[48], in stark contrast to our observations in hippocampal NGFCs. Layer I somatosensory cortical NGFCs receive excitatory input from intra-cortical sources and the ventroposteromedial/posteromedial thalamic nuclei[47]. Based on the numbers of cells that send afferents onto these INs the authors demonstrate that *Grin1* or *Grin2B* knockout during postnatal development results in a reduction in the thalamic input but is accompanied by increase in the number of presynaptic partners originating within cortical regions[47]. This loss of NMDARs also precipitates an apparent loss of dendritic arborization and complexity[47]. In the same study, cortical VIP INs are not sensitive to NMDAR-ablation with regard to their synaptic connectivity and morphological development[47]. Here we demonstrate a significant increase in both thalamic and cortical input to hippocampal NGFCs and moreover, the NGFC dendritic arbor undergoes extensive hypertrophy. Thus, even with the sparse information currently available, it is evident that the cellular and synaptic deficits following NMDAR-hypofunction can markedly vary even within the same IN subtype located in differing brain regions.

What are the functional consequences of NMDAR-hypofunction mediated abnormalities in hippocampal NGFC development? Hippocampal INs play an important role in driving a number of temporally distinct network oscillations. Although perisomatic-targeting INs, particularly those of the PV subtype, possess numerous physiological features allowing them to precisely synchronize large ensembles of PCs[49, 50], the role of dendritic-targeting IN subtypes in generating oscillations has been described[51–53]. In particular, NGFCs are phase locked to fire at the peak of each theta cycle consistent with afferent drive originating from extra-hippocampal sources such as the entorhinal cortex[54]. Recently, network computational approaches have identified NGFCs as theta pacemakers[55] and therefore must play an integral role in cognitive processes associated with such oscillations. Like the NGFC, the oriens-laconosum moleculare (O–LM) IN also target the distal dendrites of CA1 PCs[56, 57]. O–LM activity gates information flow from the mEC and within the hippocampus[58, 59] to influence expression of synaptic plasticity in PCs[58] and their recruitment is critical in learning behaviors such as fear conditioning[60]. Therefore, it is plausible that NGFCs can play a similar role to those described for O–LM INs. The NRe acts as a relay between the medial prefrontal cortex and hippocampus and their cross-talk is implicated in spatial navigation and working memory[61–63]. Taken together, the dysregulation of the long-range mEC and thalamic afferent inputs onto NGFCs described here, may result in network deficits including oscillatory activity to ultimately manifest in behavioral abnormalities.

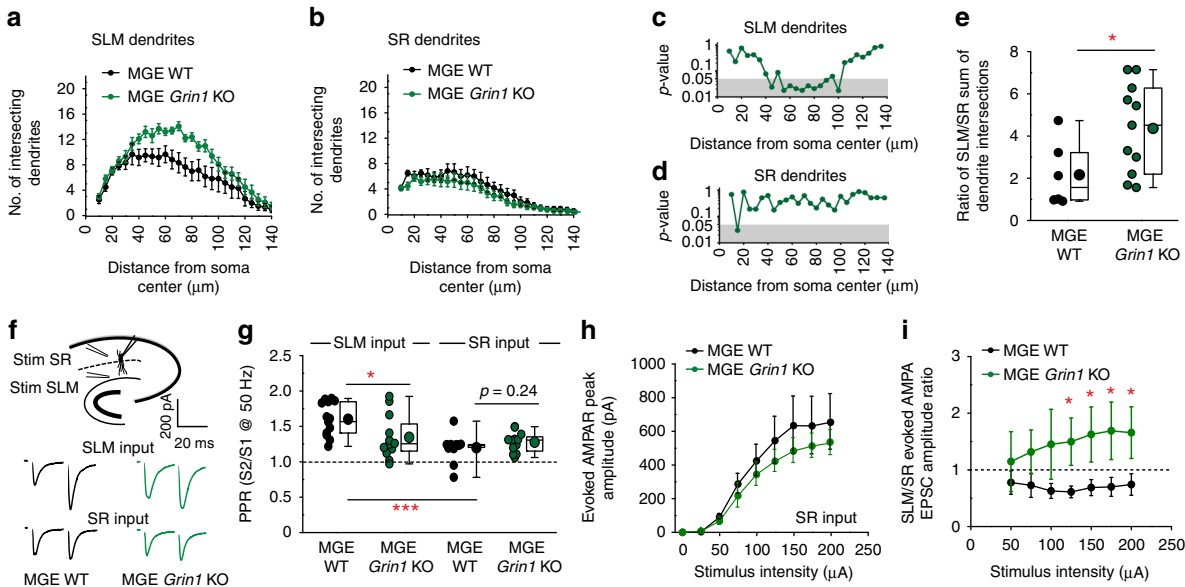

**Fig. 7** NMDAR hypofunction shifts the relative contributions of AMPA receptor mediated excitatory drive at extra- vs. intra-hippocampal inputs onto individual NGFCs. **a**, **b** Sholl analyses plots of total number of intersecting dendrites at each concentric circle (0–135 μm from center of soma every 5 μm) in WT (black data; n = 6) and Grin1 KO mice (green data, n = 11) for SLM and SR dendrites in individual MGE NGFCs. Note these data are reanalyzed from the total dendrite Sholl analyses depicted in Fig. 5g. **c**, **d** Statistical comparison of the number of intersecting NGFC dendrites at each concentric distance away from the cell soma in WT vs. Grin1 KO mice (p-values are derived from Mann–Whitney U test). Shaded gray areas denote p < 0.05. (**e**) Ratio of the sum of SLM vs. SR intersecting dendrites (between 40–100 μm away from soma center) in each individual MGE NGFC analyzed in WT (black circles; n = 5) and Grin1 KO (green circles; n = 11). (**f**) Top panel; schematic depicting the recording configuration and stimulus (stim) electrode arrangement to evoke AMPAR-mediated synaptic input mediated by SR and SLM afferents onto the same NGFC. Bottom panel; single trace examples of paired pulse ratio (PPR) at 50 Hz of SLM and SR inputs onto an individual NGFC in WT (black traces) and Grin1 KO (green traces) mice. (**g**) Pooled data of PPR in P16-21 WT (black circles; n = 13, 12 for SLM and SR input, respectively) and Grin1 KO mice (green circles; n = 12, 11 for SLM and SR input, respectively). (**h**) Relationship curves of SR evoked AMPAR-mediated EPSC amplitude vs. stimulus strength onto MGE NGFCs in WT (black data; n = 6) and Grin1 KO mice (green data; n = 7). (**i**) Ratio of the SLM vs. SR evoked maximal AMPAR-mediated EPSC amplitude vs. stimulus strength onto individual MGE NGFCs in WT (black data; n = 6) and Grin1 KO mice (green data; n = 7). Mann–Whitney U-tests were used for all comparisons (**p < 0.01, ***p < 0.001). All n values correspond to the number of cells recorded. Construction of box-whisker plots is detailed in Methods section

Recently, an unidentified IN expressing neuropeptide Y (NPY; a molecular marker for NGFCs[64–66]) residing on the CA1 SR/SLM border was shown to receive both extra-hippocampal SLM and local CA3 afferent inputs[67]. These INs impose an inhibitory filter gating the dual pathway integration of these synaptic inputs by PCs[67]. The short-term dynamics of recruitment via the long-range and local hippocampal inputs onto these INs are partly responsible for the properties of this filter[67]. It is plausible that a subpopulation of hippocampal NGFCs, by virtue of their dendritic polarization in the SLM and SR, could also serve this integrative role. Therefore, the afferent specific NMDAR-hypofunction mediated changes in synaptic strength and Pr observed in the current study could disrupt the dynamics of this inhibitory filter impacting the temporally sensitive processing of sensory input by PCs critical for synaptic plasticity required for various learning and memory processes[68].

Although descriptions of the physiological and circuit functions of NGFCs are in their relative infancy[27], we outline potential network and behavioral consequences that may result from the described abnormalities of NGFC integration revealed in the current study. Since the cre-lox strategy used here also targets a range of other IN subtypes, a direct assessment as to the consequences of NMDAR-hypofunction in NGFCs at the network/behavioral level cannot be currently undertaken. However, future development and application of cell-type specific conditional mouse lines[69] including those employing combinatorial[70] and CRISPR/Cas9 approaches to exclusively target NGFCs will undoubtedly reveal specific behavioral correlates of their recruitment and the consequences of NMDAR

hypofunction in this understudied IN subtype. Nevertheless, our study constitutes a significant contribution leading to the notion that NMDAR-signaling controls numerous developmental programs that are dependent on neuronal subtype, brain region and afferent input. Thus, it is evident a thorough and pragmatic approach in elucidating the complex array of cellular/synaptic deficits following NMDAR hypofunction is required to reveal all the potential targets for possible therapeutic intervention of neurological disorders.

## Methods

**Animals.** All experiments were conducted in accordance with animal protocols approved by the National Institutes of Health. The MMRRC/GENSAT Htr3A-EGFP (founder line DH30) and Htr3A BAC-Cre mouse lines (founder line NO152) were obtained from C. Gerfen (NIMH, NIH). The floxed Grin1 KO mouse line (B6.129S4-Grin1^tm2Stl/J; Cat. No. 005426) and Ai14 reporter mouse (B6.Cg-Gt (ROSA)26Sor^tm14(CAG-tdTomato)Hze/J; Cat. No. 007914) were purchased from the Jackson Laboratory.

**Electrophysiology.** P6-11 or P16-21 mice (for details refer to the results section) were anesthetized with isoflurane, and the brain dissected out in ice-cold saline solution containing (in mM): 130 NaCl, 25 NaHCO_3, 1.25 NaH_2PO_4, 3.5 KCl, 4.5 MgCl_2, 0.5 CaCl_2, 10 glucose, saturated with 95% O_2 and 5% CO_2 (pH 7.4). Transverse hippocampal slices (300 μm) were cut using a VT-1000S vibratome (Leica Microsystems, Bannockburn, IL, USA) and incubated in the above solution at 35 °C for recovery (1 h), after which they were kept at room temperature until use. Individual slices were transferred to an upright Olympus BX51WI microscope and visualized with infrared differential interference contrast microscopy. Slices were perfused (2 ml/min) with extracellular solution composed of (in mM) 130 NaCl, 24 NaHCO_3, 3.5 KCl, 1.25 NaH_2PO_4, 2.5 CaCl_2, 1.5 MgCl_2, and 10 glucose, saturated with 95% O_2 and 5% CO_2 (pH 7.4). The extracellular solution was routinely supplemented with 50 μM picrotoxin and 2–5 μM CGP 55845A

(Tocris Science or Ascent Scientific). Recordings were performed at 32–34 °C with electrodes (3–5 MΩ) pulled from borosilicate glass (World Precision Instruments, Sarasota, FL) filled with one of two intracellular solutions (in mM); (i) 150 K–gluconate, 3 MgCl$_2$, 0.5 EGTA, 2 MgATP, 0.3 Na$_2$GTP, 10 HEPES and 3 mg/ml biocytin; (ii) 135 CsMeSO$_4$, 8 KCl, 4 MgATP, 0.3 Na$_2$GTP, 5 QX–314, 0.1 spermine, 0.5 EGTA and 3 mg/ml biocytin. The pH was adjusted to 7.3 with KOH or CsOH and osmolarity adjusted to 280–300 mOsm. Whole-cell patch-clamp recordings were made using a Multiclamp 700B amplifier (Molecular Devices, Sunnyvale, CA, USA). Signals were filtered at 4–10 kHz and digitized at 10–20 kHz (Digidata 1322A and pClamp 9.2 or 10.2 Software; Molecular Devices, Sunnyvale, CA, USA). Glutamate receptor mediated synaptic responses were evoked with A360 constant-current stimulus isolator (World Precision Instruments, Sarasota, FL, USA) and with stimulation electrodes pulled from borosilicate glass filled with aCSF placed in SLM and/or SR. In experiments where both SLM and SR inputs were evoked onto a single NGFC we confirmed that non-overlapping afferents were indeed activated since a single SR stimulus following the SLM stimulus or vice versa resulted in no short-term plasticity and did not recapitulate the paired pulse facilitation noted upon stimulation of either input alone (Fig. 7f, g; SR input following SLM input, PPR = 106 ± 4; SLM input following SR input, PPR = 100 ± 5; $n$ = 5). To measure NMDAR/AMPAR (NA) ratios cells were initially voltage-clamped (Vh = −70mV) and AMPAR-mediated EPSCs evoked. Following addition of 5 µM DNQX (Tocris Science or Ascent Scientific), cells were held at Vh = + 50 mV and NMDAR-mediated EPSCs evoked using the exact same stimulus parameters and NMDAR kinetics expressed as a weighted time constant calculated from a two site exponential decay fit. In a subset of experiments, 100–300 µM DL-AP5 (Tocris Science or Ascent Scientific) was added to confirm the NMDAR identity of the EPSC at + 50 mV. To measure sEPSC$_{AMPA}$ events, cells were held at Vh-70 mV and gap-free recordings for 30–90 s were performed. Automation of event detection was achieved using a template matching protocol (Clampfit) and sEPSC$_{AMPA}$ frequency was calculated. sEPSC$_{AMPA}$ amplitudes were measured for each individual event and mean values calculated. In addition, 30 events from each cell were randomly picked and pooled together (from all cells within a dataset) to construct cumulative distribution curves of sEPSC$_{AMPA}$ amplitudes.

For determination of membrane and firing parameter membrane potential was biased to −60 mV by constant-current injection. To determine neuronal resting membrane potential without disrupting the cell's intracellular environment, we monitored potassium-channel activation during depolarizing voltage ramps (from −100 to + 200 mV) applied to cell-attached patches prior to breakthrough into the whole-cell configuration. After breakthrough, input resistance ($R_m$) was measured using a linear regression of voltage deflections (±15 mV from resting potential, ~60 mV) in response to 2 s current steps (increment 5 pA). Membrane time constant ($\tau_m$) was calculated from the mean responses to 20 successive hyperpolarizing current pulses (−20 pA; 400 ms) and was determined by fitting voltage responses with a single exponential function. AP generation was achieved by employing incrementing steps of positive current injections (800 ms). The rheobase was defined as the minimum current injection to elicit AP firing (i.e., threshold). Late-spiking phenotype was assessed/quantified as the delay to the first AP peak measured at threshold current injection. Firing frequency was calculated from the number of spikes observed at 2× threshold. To determine the sag index of each cell we used a series of 800 ms negative current steps to create V–I plots of the peak negative voltage deflection ($V_{hyp}$) and the steady state voltage deflection (average voltage over the last 200 ms of the current step; $V_{sag}$) and used the ratio of $V_{rest}$−$V_{sag}$/$V_{rest}$−$V_{hyp}$ for current injections corresponding to $V_{sag}$ = −80mV determined from polynomial fits of the V–I plots. All electrophysiological parameters were measured in pClamp or using procedures written in Igor 6 (Wavemetrics, Portland, OR, USA).

Slices containing biocytin-filled cells were drop fixed in 4% paraformaldehyde overnight at 4 °C then triton permeabilized and incubated with Alexa–488, 555 or 633 conjugated avidin (Molecular Probes, Eugene, OR, USA). Following multiple washes slices were re-sectioned (70 µm) on a freezing microtome (Microm) and mounted on gelatin-coated slides using Mowiol (EMD Millipore, Billerica, MA, USA) mounting medium. Fluorescent stacked Z-section confocal images of recorded cells revealed by biocytin conjugation were captured using an LSM 710 or 780 inverted scanning confocal instrument mounted on a Zeiss Axiovert 200M microscope (NICHD Microscope and Imaging core). For determination of dendrite complexity only the biocytin-filled NGFCs that were deemed high quality and expressed dendrites in a single re-sectioned slice of 70 µM were selected (57 NGFCs in total) for Sholl analyses. Sholl analyses were performed using the Simple Neurite Tracer plugin for Fiji (http://fiji.sc/Welcome). Concentric circles starting 10 µm from the center of soma, each 5 µm apart, were constructed. Number of intersecting dendrites for each concentric circle was measured and plotted against radial distance from the center of the soma. In the current study, 325 putative CGE and MGE NGFCs were targeted for electrophysiological recordings (including those analyzed in the optogenetic experiments detailed below) of which 278 were identified as NGFCS. Of these, biocytin recovery was deemed of sufficient quality in 53 NGFCs to perform Sholl analyses as outlined above.

**Optogenetics.** WT or Grin1 KO mice (p19–p21) were anesthetized with 5% isoflurane and mounted in a stereotax (David Kopf Instruments Model 1900, Tujunga, CA, USA) for injection of viral vectors. Topical lidocaine/prilocaine

cream (2.5%/2.5%) and buprenorphine (0.1 mg/kg via subcutaneous injection) were provided for post-operative analgesia. Mice were provided with topical lidocaine and ketoprofen daily for at least 3 days following surgery. Htr3A-EGFP mice were injected with rAAV9-Syn-ChrimsonR-tdTomato (4.3 × 10e12 virus molecules/ml, UNC Vector Core, contributed by Edward Boyden) and Htr3A-cre: Ai14:Grin1$^{flox/flox}$ mice were injected with rAAV9-Syn-Chronos-GFP (3.1 × 10e13 virus molecules/ml, UNC Vector Core, contributed by Edward Boyden). Virus was delivered via a glass micropipette attached to a syringe (Hamilton Company Inc., Reno, NV, USA) and back filled with light mineral oil. Medial enthorinal cortex (left or right hemisphere) was targeted using the following coordinates: 4.4 mm caudal and 3.3 mm lateral to bregma, and 2.6 mm deep from the pia (Supplementary Fig. 6). NRes of the thalamus was targeted using the following coordinates: 0.6 mm caudal and 0.0 mm lateral to bregma, and 4.0 mm deep from the pia (Supplementary Fig. 6). At both injection sites, 400 nl of viral vector were injected at 100 nl/min, and the pipette was left in place for 5 min following the injection before removal. After a minimum of 2 weeks post injection transverse hippocampal sections were cut in a sucrose-based saline solution compose of (in mM): 90 sucrose, 80 NaCl, 25 NaHCO$_3$, 1.25 NaH2PO$_4$, 3.5 KCl, 4.5 MgCl$_2$, 0.5 CaCl$_2$, 10 glucose, saturated with 95% O$_2$ and 5% CO$_2$ (pH 7.4). Electrophysiological recording were essentially performed as described above with the exception of the methodology for evoking glutamate receptor mediated synaptic responses. Light stimuli of 1 ms duration were delivered to the slices through the 40× water immersion objective using a CoolLED pE-4000 Illumination system (Andover, UK). Light stimulation parameters were set to attain the maximal AMPAR-mediated EPSC peak amplitude which was typically 470 and 550 nm (100% LED power) for rAAV9-Syn-Chronos-GFP and rAAV9-Syn-ChrimsonR-tdTomato, respectively. NA ratios were ascertained as previously described.

**Statistical analyses.** Non-parametric Mann–Whitney $U$-tests or Wilcoxon tests were employed for unpaired and paired data, respectively. Box-whisker plots were constructed as follows: symbol denotes mean value; line represents the median value; lower and upper box limits represent 25th and 75th percentiles, respectively. Lower and upper whiskers represent 5th and 95th percentiles, respectively.

**Data availability.** Data available on request from the authors.

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

## Acknowledgements
We thank Daniel Abebe for mouse colony maintenance and Kurt Auville for additional assistance with confocal imaging. We thank UNC Vector Core and Ed Boyden, MIT, Cambridge, MA, USA for generously providing AAV9-syn-Chrimson-TdTomato and AAV9-syn-Chronos-GFP. This work was supported by an intramural award to C.J.M. from the Eunice Kennedy–Shriver National Institute of Child Health and Human Development and a Competitive Fellowship Award to J.C.W. from the National Institute of Neurological Disorders and Stroke

## Author contributions
R.C. performed the electrophysiology experiments and Sholl analyses. J.C.W. and M.T.C. optimized and performed the channel rhodopsin viral injections. E.B., X.Y., C.F., D.C., and S.H. provided technical assistance with confocal imaging and cell recoveries. R.C. conceived the project/wrote the manuscript and R.C., K.A.P, and C.J.M designed the experiments.

## Additional information

**Competing interests:** The authors declare no competing financial interests.

