## [Peer Review File · Nature Communications]

Reviewers' comments:

Reviewer #1 (Remarks to the Author):

This paper by Chittajallu and colleagues employs an impressive array of techniques including slice electrophysiology, cell-specific genetic manipulation, genetic fate mapping and optogenetics to characterize the impact of NMDA receptor hypofunction on the integration of a subtype of hippocampal interneurons into functional networks during development. As a result of this multiplicity of techniques and because different developmental time points are analyzed, the amount of data gathered here is really remarkable and fuels eight busy figures, nicely illustrating the experiments. As rightfully summarized in the discussion section by the authors themselves, I am convinced that this study provides a significant contribution to the idea that "NMDA-R signaling controls numerous developmental programs that are dependent on neuronal subtype, brain region and afferent input". That said, I think that the paper can be significantly improved precisely by focusing its scope onto this main point (I think some of the data presented here deserves a different paper) and by further strengthening the data supporting it. Data unrelated to this main point should be removed because it blurs the main message.

The authors show that a subpopulation of a subtype of GABAergic neurons, the neurogliaform cells (NGFCs), with a soma located in the SLM and a CGE origin, displays an unusually large postsynaptic NMDA-R component, which, if selectively affected, induces cell loss and alters the development of AMPA-R mediated synaptic transmission and of dendritic processes that occurs between P6-11 and P21. In addition, it is shown that this is a cell-autonomous process, which is, in my opinion a very important finding and a crucial point for the interpretation of the data. This unusually large NMDA component, selectively expressed in interneurons and driven by thalamic and entorhinal inputs, has major implications for the functional organization of the hippocampus. In conclusion, if the issues detailed below are addressed, this elegant study includes many important findings both at developmental and functional levels, with interesting implications for the understanding of brain disorders related to hippocampal dysfunction.

Major issues

1) Cell specificity: it is difficult to understand how specific is the neuronal subtype studied here (i.e. the subtype of neuron for which a large NMDA-R component as a developmental function), and if specific, which criteria specify it: is it the NGFCs (both CGE and MGE derived), the neurons with a dendritic arbor in SLM, the CGE-derived NGFCs, the CGE-derived SR/SLM neurons, etc? In other words, is this subtype specified by developmental origin by morphology and neurochemical content or just by soma location? Hence we do not know whether the high N/A ratio is specific to CGE-derived NGFCs (as expected from the conclusions of a previous publication by the same group, Matta et al. NN 2013), or whether it also applies to MGE-derived NGFCs which are briefly analyzed at the end of the manuscript. If morphological identification of NGFCs is the major criterion to specify the subtype, then the manuscript should include the number of cells for which anatomical identification was possible and out of those, the total number of reconstructed neurons. Regarding the same issue, I think the introductory paragraph is a little misleading in presenting the subtype of cell studied here. Indeed, it states (lines 64-65) that the focus is on neurogliaform cells. Indeed, the paper is mostly focusing on CGE-derived NGFCs which, according to Tricoire et al. 2010 are a small minority of nNOS negative NGFCs. In addition, with the Ai14 reporter it is a subpopulation of that subset of NGFCs that is labelled. I think the introduction should at least introduce what is known, from the previous work of the authors, about the embryonic origin of NGFCs.

2) Developmental issue: This manuscript focuses on the developmental role of the NMDA component in the functional integration of CGE-derived NGFCs between P6-11 and P15-21. I Therefore think that

in this case, it would be better to limit and strengthen the observations related to changes in parameters that are developmentally regulated. In this respect, I would remove the description of the GluA2 contribution to the glutamatergic response in these cells since there is no developmental change of GluA2-R contribution for that cell type, so the rationale for those experiments is difficult to follow. In fact the conclusion of this part of the MS is too weak (lines 340-344) to justify this analysis. Rather, I think it would be interesting to know whether GABAergic innervation of these cells is also affected, since the phenomenon described here is shown to be cell autonomous. Also, the change in morphology of the dendritic arbor observed when NMDA-R is hypofunctional, does it recapitulate a developmental process (i.e. the cells recorded at P6-11 and reconstructed, do they display a more widespread dendritic arbor similar to the P16-21 Grin1KO ones?). Last, from the point of view of development, it would have been interesting to know how these cells functionally integrate adult networks by providing some experiments in adult slices, but I understand that the MS already contains a lot of information, so this is not absolutely necessary.

3) Focus issue: this paper gathers an impressive amount of data and the authors should clearly be acknowledged for that effort. However, the drawback is that the MS in its present form is quite heavy to read and this, even more since the rationale for some experiments is not always straightforward. I suggest removing the GluA2 experiments. I am not either convinced about the reelin experiments. Indeed, I did not understand why the authors analyzed reelin expression rather than the more classical markers for NGCs (NPY and nNOS). From Tricoire et al. 2011, about 60% of the MGE-derived Ins in LM express reelin. In the present experiments there seems to be no more reelin expression in SLM (in Fig. 2d, but the quantification in Fig. 2h is different) in the CGE-Grin1KO, could it be that MGE-derived NGCs of the LM are also affected? I did not understand how the authors could estimate a cell density per mm³ and why not quantify the density per mm²?

Minor issues:

Fig. 1g: please explain what dashed line indicates

Supp. Fig. 1: cell counts are done in slices? I am not sure that "hippocampi" is the correct wording since one mouse cannot contain 8-10 hippocampi (same for Fig. 2 legend)!

Fig. 1b: post hoc reconstruction was performed in how many EGFP cells and out of those what is the fraction of cells displaying a NGFC morphology?

Line 141: please introduce what experiments are being done next

Supp. Fig. 2: is this done using the Ai14 mouse line?

Lines 259-260: a correlation cannot "causally link" two observations, please rephrase.

P. 10 line 304: "causal relationship" may not be the appropriate wording

-The discussion could emphasize better the "cell autonomous" effect of NMDA-R hypofunction and maybe speculate a little bit on the molecular pathways linking NMDA-R hypofunction and AMPA-R development?

Reviewer #2 (Remarks to the Author):

The study by Chittajallu R et al. tackles the role of NMDA receptors in the developmental integration of neurogliaform cells of the Stratum Lacunosum Moleculare (S-LM) in the CA1 area of hippocampus.

This study comes as a follow up to previously published work by the same group on the development of Medial and Caudal Ganglionic Eminence- derived interneurons populating the CA1 area. Here, the authors tackle the role of activity through NMDA receptors in the development of the intrinsic and synaptic electrophysiological properties of a specific type of interneuron, called neurogliaform cell (NGFC). The study is timely and overall well performed by a group with significant experience in analysis of synaptic properties on and by hippocampal GABAergic interneurons.

The authors report a number of alterations after removal of NMDA receptors from NGFCs during development, which are presented in a sequential and somewhat incremental manner. Although the study presents some novel findings, there are a number of studies that have assessed the role of activity in the development of cortical neurons, both pyramidal and interneurons, including neurogliaform cells, using more or less the same toolkit as the authors herein (most of them are cited in this manuscript). Importantly, the study seems to lack a core thread and has rather been laid out as a series of analyses after knocking out the Grin1 subunit of NMDA receptors. A major re-write would facilitate in bringing up and leading to what the reviewer sees as the key message of this paper, namely the TC versus thalamic input differences between WT and KO cells.

More specifically:

Chittajallu et al. report that NGFCs of the CA1 hippocampal area have the largest NMDA/AMPA ratio observed in any of the interneurons studied so far, the mean of which is around "3". This is largely based on the work of the authors themselves in a previously published study (Matta J et al. 2014), in which they recorded from a number of MGE- and CGE-derived hippocampal interneurons, but not SLM ones, which include NGFCs. In that study the authors report that the highest ratios are found in CGE-derived cells, some of which have a value of "2". Upon a closer look of the data presented in the current paper, there is a widespread distribution of ratio values ranging from 1 to 5.5 in the 16-21 age group. This means that there is an overlap between the ratios of different CGE-derived cells and the values likely form a continuum. Have the authors done a statistical comparison between the distributions of other CGE-derived cells and NGFCs to see if they are different?

In lines 121-123 the authors write "Therefore, Htr3A-promoter driven cre-expression is apparent from the CGE-progenitor stage but is only evident in a subpopulation of CGE-derived NGFCs demonstrating a mosaic pattern of cre-expression." It is not clear why the authors claim that all the GFP-positive cells are NGFCs since the only recorded from them in the first set of experiments and subsequently worked on and characterized the Cre-Ai14 fate-mapped cells. Both mouse lines that the authors are using here are BAC transgenics and unless one performs double in situ hybridization analysis against the fluorophores and the actual 5-HT3A mRNA it is not easy to make such a statement. It could be, that the 5-HT3A-GFP mouse line also marks some Cajal-Retzius cells found in the SLM of CA1.

In the section under "Lamina specific reduction in IN density and reelin-expression" the authors report that the only population that they see a reduction of is the NGFCs positioned in the SLM, in striking contrast to cells that are found in other layers, in the populated Stratum Oriens (SO), but also in the less so Stratum Radiatum (SR). In light of the partial penetrance of NMDA receptor removal, have the authors performed electrophysiological recordings to test if the NMDA currents are absent in the cells in SO and SR so as to strengthen their claim about lamina specificity?

The genetic tools are well chosen, although it is not clear why the authors see some cells with a complete loss of NMDA receptors and others with only partial or none. Is the "Cre" not strong enough in some cells or has it not fully turned on at the stage when the experiments were done? Have the authors ever performed a double in situ hybridization protocol to test for the expression of Cre mRNA in the GFP+ cells? It would also be helpful if the authors discussed how this partial removal may relate to the reduction in cell number the authors observe in SLM, also as compared to the results obtained by Kelsch W et al. 2014, where they show that removal of the NR2B subunit from all interneurons does not affect their numbers or distribution in the cortex.

The authors find that upon removal of NMDA receptors the frequency of spontaneous AMPA receptor-mediated currents increases, as does the evoked current recorded after electrically stimulating in the same layer. At the same time the dendritic length of the cells increases and the paired-pulse ratio

(PPR) at P16-21 decreases. How are these findings related? And what do the authors think is the primary effect of removing NMDA receptors versus secondary?

Could the changes in the observed PPR be the result of postsynaptic rather than presynaptic changes? The authors stimulate at a fairly high frequency, 50Hz, leaving the possibility open that the synaptic depression observed in the KO is partially due to receptor desensitization. Has a lower frequency of stimulation been tried?

The optogenetic experiments and results are very interesting.

The authors show nicely that there are terminals labeled in the SLM of CA1 after injections in either the mEC or NRe. It would be valuable if they showed how the expression looks like at the site of injection for each of the two spots.

The authors report that only after light-stimulation of the mEC inputs, but not the NRe, in the absence of NMDA receptors they obtain a statistically-significant increase in the maximum AMPA receptor-mediated synaptic current evoked. Although there seems to be a specific increase in the mEC inputs, the variability of the evoked responses in the absence of Grin1 is quite large and with a data point that appears to be an outlier. If the authors excluded that point would they still get statistically-different responses compared to WT?

Also, the reviewer was wondering how do the authors potentially relate the specificity obtained in the optogenetic experiments with the changes observed in the morphology of the cells?

Methodologically, it seems that these optogenetic experiments were not performed in the presence of TTX and 4AP, as reported in previous studies (for ex. Mao T et al. Neuron 2011). Is it hence plausible that part of the current recorded does not come from the direct inputs from the mEC or NRe onto the neurogliaform cells, but rather through intermediate cells?

Reviewer #3 (Remarks to the Author):

The study by Chittajallu et al. 'NMDA receptor hypofunction disrupts the developmental programs critical for appropriate circuit integration of neurogliaform cells' describes the developmental trajectory of glutamate synapse maturation in a genetically and functionally identified interneuron type in the postnatal hippocampus. The authors demonstrate with state-of-the-art methods and analyses the developmental consequences of NMDA hypofunction or loss-of-function on spontaneous synaptic inputs and presynaptic features. They exploit elegantly conditional mutant lines to generate mosaic (partial and complete) gene loss-of-function and demonstrate that the changes in synaptic development occur in a cell autonomous fashion. The paper also convincingly demonstrates that NMDAR loss-of-function differentially alters inputs to genetically-defined subsets of this interneuron type. The study thus substantially advances our view of the heterogeneity of developmental NMDAR function and will influence thinking in the field. A few points are outlined below that still need to be improved.

Major points

1. Did the authors test whether the NMDAR hypofunction observed at P16-21 would eventually switch after P21 to a bimodal distribution in dTom+ cells with either no or normal NMDAR component? In other words, is hypofunction just a transient phenomenon? This appears critical to understand the logic of the recombination event. Further, a more detailed discussion of Cre expression levels and developmental time course appear critical for understanding the results.

2. At which time point is the reduced number of dTom+ cells in GluN1 ko mice first observed (related to Fig. 2a,b,g). Here further evidence needs to be provided whether cells are lost during maturation after they took their eventual position and/or already display impaired migration.

3. Were reconstructions made from 50 μm thick slices as described in the histology section? Please provide here further technical details. If indeed 50 μm sections were used and considering the indicated span of dendrites $>200 \mu\text{m}$, truncation is expected to be substantial. Changes in morphological feature of total dendritic length vs branching may be hard to assess. Also, it is not clear in how far synapse densities correlates to dendrite surface. Considering these points, the authors may moderate the conclusions and add Fig. 5 e-I as supporting evidence to the Suppl. Mat. or provide stronger evidence for correlations of structural synapse densities and dendritic length in wt and mutant cells. The same question essentially applies to Fig. 8e-I where the larger SLM dendrites are more prone to the above mentioned analysis problems.

4. Discussion: Except for the detailed difference between the present study and the two previous studies on NMDAR deletion in INs, the ms. would benefit from a more focused and critical discussion of their own findings e.g. cell autonomous effects and potential relation of the different changes in synapse function. Also, a more careful discussion of NGFCs and their relation to other interneuron types in disease context would strengthen the ms. In particular, evidence of dendrite vs. soma-targeting INs in pathology may be disentangled as far as possible.

Minor points

1. Please define MGE/CGE when first used.

2. Please provide technical details how cells were quantified in Suppl. Fig 1. Please check whether 'hippocampi' actually should say 'section' in the Fig. legend: '8-10 hippocampi counted per mouse; number of cells counted per hippocampus = 7 - 21). Was the entire SLM imaged or only part of it in each section?

3. Green lines are hard to distinguish in Fig. 1o.

4. Related to Fig. 3d, p-values should be provided in the text for the comparison between WT vs. Cre- in grin1 ko and NA<1 vs Cre- in grin1 ko cells at P16-20 to support conclusions.

5. The 'causal' relationship is well demonstrated between the loss of GluN1 and PPR or EPSC freq changes, respect. in a cell autonomous fashion, but are PPR changes necessarily 'causal' to sEPSC freq changes as claimed in l. 260? Or could two effects co-exist and for instance (functional) synapse densities differ additionally between mutant and wt cells?

Reviewers' comments:

Reviewer #1 (Remarks to the Author):

This paper by Chittajallu and colleagues employs an impressive array of techniques including slice electrophysiology, cell-specific genetic manipulation, genetic fate mapping and optogenetics to characterize the impact of NMDA receptor hypofunction on the integration of a subtype of hippocampal interneurons into functional networks during development. As a result of this multiplicity of techniques and because different developmental time points are analyzed, the amount of data gathered here is really remarkable and fuels eight busy figures, nicely illustrating the experiments. As rightfully summarized in the discussion section by the authors themselves, I am convinced that this study provides a significant contribution to the idea that “NMDA-R signaling controls numerous developmental programs that are dependent on neuronal subtype, brain region and afferent input”. That said, I think that the paper can be significantly improved precisely by focusing its scope onto this main point (I think some of the data presented here deserves a different paper) and by further strengthening the data supporting it. Data unrelated to this main point should be removed because it blurs the main message. The authors show that a subpopulation of a subtype of GABAergic neurons, the neurogliaform cells (NGFCs), with a soma located in the SLM and a CGE origin, displays an unusually large postsynaptic NMDA-R component, which, if selectively affected, induces cell loss and alters the development of AMPA-R mediated synaptic transmission and of dendritic processes that occurs between P6-11 and P21. In addition, it is shown that this is a cell-autonomous process, which is, in my opinion a very important finding and a crucial point for the interpretation of the data. This unusually large NMDA component, selectively expressed in interneurons and driven by thalamic and entorhinal inputs, has major implications for the functional organization of the hippocampus. In conclusion, if the issues detailed below are addressed, this elegant study includes many important findings both at developmental and functional levels, with interesting implications for the understanding of brain disorders related to hippocampal dysfunction.

We thank the reviewer for their enthusiasm for our study. We agree with his/her comments pertaining to issues concerning the overall focus and framing of the study. Accordingly, we have significantly revised the manuscript. Major changes include removal of removal of the GluA2 and cell counting data, as the Reviewer suggests. We have also incorporated additional datasets that include morphological analyses of P6-11 NGFCs and further channelrhodopsin experiments dissecting the extra-hippocampal inputs onto NGFCs. The revised manuscript now clearly consists of the first part relating to our observations regarding the deficits of synaptic function during postnatal development following NMDAR ablation and the second part dissecting the various excitatory inputs that impinge on hippocampal NGFCs resulting in the conclusion that morpho-physiological properties are differentially affected dependent on afferent identity. We once again thank the Reviewer for the very helpful comments that we believe have improved our manuscript.

Major issues

1. Cell specificity: it is difficult to understand how specific is the neuronal subtype studied here (i.e. the subtype of neuron for which a large NMDA-R component as a developmental function),

and if specific, which criteria specify it: is it the NGFCs (both CGE and MGE derived), the neurons with a dendritic arbor in SLM, the CGE-derived NGFCs, the CGE-derived SR/SLM neurons, etc? In other words, is this subtype specified by developmental origin by morphology and neurochemical content or just by soma location? Hence we do not know whether the high N/A ratio is specific to CGE-derived NGFCs (as expected from the conclusions of a previous publication by the same group, Matta et al. NN 2013), or whether it also applies to MGE-derived NGFCs which are briefly analyzed at the end of the manuscript. If morphological identification of NGFCs is the major criterion to specify the subtype, then the manuscript should include the number of cells for which anatomical identification was possible and out of those, the total number of reconstructed neurons.

We are definitely cognizant of the potential difficulties in conducting a study focused on a particular subtype of interneuron particularly across developmental periods. With particular regard to NGFCs, their targeting for study is somewhat easier due to their distinct small soma size and their relative abundance in the regions in which we perform our recordings i.e. for CGE NGFCs we focus on soma located in the superficial SLM regions of CA1. For the MGE NGFCs we point out in the manuscript that very rarely do we see NKX2.1 reported cells in the superficial SLM (**Fig. 5a**), instead they are predominantly located on the border of SLM/SR. We should point out that the major MGE-derived hippocampal interneuron subtypes reported in the NKX2.1-cre mouse are PV-expressing, SOM-expressing and a subset of NGFCs termed Ivy cells. The cell bodies of these MGE interneurons are primarily restricted to deep SR, PC layer and SO. Over many experiments, as the reviewer can appreciate from the large numbers of n's in the final manuscript (data from a total of 278 NGFCs), we are relatively proficient at using the soma size, shape and orientation of proximal portions of their dendrites which could be also be visualized in the reporter mice to target putative NGFCs. Of course, we additionally employed morphological confirmation using criteria including relatively small dendrites (<140 μ M) and dense axonal arborizations that outspan these small dendritic fields. The cells not included in the datasets comprise those in which biocytin recovery was not sufficient to identify the cell or the morphology did not fit that reminiscent of NGFCs (47 cells in total) and thus we achieved a high percentage of success (~ 85%) in targeting both CGE and MGE NGFCs using the reporter mice, soma size and lamina position. The use of a subset of identified NGFCs for Sholl analyses (53 NGFCs in total) was due to only selecting those NGFCs with the highest quality of biocytin staining and also where their dendritic arbor was confined to a single 70 μ M re-sectioned slice (also please refer to response to Reviewer 3; Point 3). As the reviewer requested we have now included in the methodology the number of cells that we identify as NGFCs out of the total number of cells recorded for both the electrophysiological and subsequent morphological Sholl analyses (lines 691-695). We have also included morphology of a typical P6-11 NGFCs to illustrate the

With regard the comment on the developmental origin the Reviewer is correct to point out that we have previously shown that hippocampal CGE interneurons possess higher NA ratio than their MGE counterparts (~1 versus 0.3). In the current manuscript we show that both CGE and MGE NGFCs possess NA ratios of approximately 3 and 1.5, respectively. Thus, regardless of embryonic origin NGFCs possess higher NA ratios than the other CGE and MGE IN subtypes previously analyzed in the Matta *et al.* study. We have now included a direct graphical comparison of NA ratios in CGE versus MGE NGFC populations resulting from both electrical and channel rhodopsin afferent stimulation techniques (**Fig. 5c and Fig. 6e,f; lines 359-366**).

2. Regarding the same issue, I think the introductory paragraph is a little misleading in presenting the subtype of cell studied here. Indeed, it states (lines 64-65) that the focus is on neurogliaform cells. Indeed, the paper is mostly focusing on CGE-derived NGFCs which, according to Tricoire et al. 2010 are a small minority of nNOS negative NGFCs. In addition, with the Ai14 reporter it is a subpopulation of that subset of NGFCs that is labelled. I think the introduction should at least introduce what is known, from the previous work of the authors, about the embryonic origin of NGFCs.

The ability to label distinct interneuron subtypes can be highly dependent on the transgenic reporter mouse line employed. In our study, and as the Reviewer correctly points out, the 5HT3ARCre: Ai14 mouse only labels approximately 40-50% of the total putative CGE NGFCs reported in the more comprehensive 5HT3AR-EGFP line. In the Tricoire *et al* paper we ascertained that MGE NGFCs expressed nNOS whereas the majority of CGE NGFCs did not. However, we employed the Mash1CreER mouse line to label CGE interneurons including those of the NGFCs subtype. In this line, Cre activity and hence fluorescent reporting requires tamoxifen induction that in this previous study was administered on a single day of embryonic development resulting in labeling of a small percentage of CGE NGFCs. In fact, CGE-derived NGFCs are relatively abundant and likely comprise the largest subset of NGFCs within the hippocampus. We hope this clarifies this point but nevertheless we agree with the reviewer that and have included a sentence in the introduction to inform readers as to the existence of dual origins of hippocampal NGFCs (lines 72-74).

3. Developmental issue: This manuscript focuses on the developmental role of the NMDA component in the functional integration of CGE-derived NGFCs between P6-11 and P15-21. I therefore think that in this case, it would be better to limit and strengthen the observations related to changes in parameters that are developmentally regulated. In this respect, I would remove the description of the GluA2 contribution to the glutamatergic response in these cells since there is no developmental change of GluA2-R contribution for that cell type, so the rationale for those experiments is difficult to follow. In fact the conclusion of this part of the MS is too weak (lines 340-344) to justify this analysis. Rather, I think it would be interesting to know whether GABAergic innervation of these cells is also affected, since the phenomenon described here is shown to be cell autonomous.

We agree and have now removed all discussion of experiments relating to GluA2. Regarding the inhibitory “output” experiments we agree with the reviewer that this is an extremely interesting avenue of investigation, especially in light of recent observations demonstrating a role of NMDAR function in the development of inhibitory input onto pyramidal cells (Gu et al., Cell Reports 14, 471-478; 2016). However, we feel that this constitutes a separate study and therefore refrained from pursuing this path in this current manuscript and thank the Reviewer for his/her suggestion.

4. Also, the change in morphology of the dendritic arbor observed when NMDA-R is hypofunctional, does it recapitulate a developmental process (i.e. the cells recorded at P6-11 and reconstructed, do they display a more widespread dendritic arbor similar to the P16-21 Grin1KO ones?).

We have now performed additional Sholl analyses in the “young” P6-11 NGFCs and the new

data are found in **Fig. 4**. Our data demonstrates that dendrite complexity under a normal developmental scenario increases between P6-11 and P16-21 (**Fig. 1c,d and Fig. 4a-c**). The addition of these analyses clearly demonstrates that the dendrite exuberance precipitated by NMDAR hypofunction is not likely via a deficit in dendrite pruning. Furthermore, the data allows us to hypothesize about the relative contributing factors resulting in the increase of sEPSC frequency caused by distinct timing of changes in dendrite complexity and PPR across the development ages tested (**lines 306-332**).

5. Last, from the point of view of development, it would have been interesting to know how these cells functionally integrate adult networks by providing some experiments in adult slices, but I understand that the MS already contains a lot of information, so this is not absolutely necessary.

We should point out that the channel rhodopsin experiments were performed between p35-p57. We also performed gap-free recordings in a subset of the NGFCs in which light evoked EPSCs were measured. We show at this age a significant increase in sEPSC AMPA frequency in *Grin1* KO mice. (sEPSC frequency in NGFCs from control vs *Grin1* KO mice = 1.6 ± 0.14 and 7.3 ± 0.9 ; n=10 and 15, respectively). This data is now included in the text of the manuscript (**lines 426-430**). Thus, although we did not undertake a full analysis of all morpho-physiological parameters tested at the earlier ages the increase in sEPSC_{AMPA} frequency and maximal light-evoked AMPAR EPSC amplitude persists till adult stages.

6. Focus issue: this paper gathers an impressive amount of data and the authors should clearly be acknowledged for that effort. However, the drawback is that the MS in its present form is quite heavy to read and this, even more since the rationale for some experiments is not always straightforward. I suggest removing the GluA2 experiments. I am not either convinced about the reelin experiments. Indeed, I did not understand why the authors analyzed reelin expression rather than the more classical markers for NGCs (NPY and nNOS). From Tricoire et al. 2011, about 60% of the MGE-derived Ins in LM express reelin. In the present experiments there seems to be no more reelin expression in SLM (in Fig. 2d, but the quantification in Fig. 2h is different) in the CGE-*Grin1*KO, could it be that MGE-derived NGCs of the LM are also affected? I did not understand how the authors could estimate a cell density per mm³ and why not quantify the density per mm²?

As mentioned earlier we have removed the data pertaining to GluA2 expression. Additionally the cell counting data, although of interest, disrupts the flow of the manuscript and detracts from the main focus of the paper that is predominantly centered on the synaptic and functional aspects of NGFC development following NMDAR ablation. Following the concerns of the reviewer we have also eliminated discussion of reelin expression.

Minor issues:

Fig. 1g: please explain what dashed line indicates

The dashed line corresponds to an NA ratio of 1 to demonstrate that virtually all NGFCs in WT have an NA ratio greater than this value. This is important since in many instances we

subsequently parse our data from *Grin1* KO NGFCs that express either NA<1 or NA>1 to analyze those cells, which have measurable hypofunction versus those with no appreciable NMDAR ablation. We have now indicated the significance of the dotted line in figure legend.

Supp. Fig. 1: cell counts are done in slices? I am not sure that “hippocampi” is the correct wording since one mouse cannot contain 8-10 hippocampi (same for Fig. 2 legend)!

We apologize for the confusion and this should read “8-10 hippocampal sections” and have now amended the legend accordingly (Note: these data are now in **Supp. Fig. 3** of the revised ms).

Note: the counting data in the old Figure 2 has now been removed.

Fig. 1b: post hoc reconstruction was performed in how many EGFP cells and out of those what is the fraction of cells displaying a NGFC morphology?

As outlined in Response 1 we have now included these values in the methods sections (**lines 691-695**).

Line 141: please introduce what experiments are being done next

Supp. Fig. 2: is this done using the Ai14 mouse line?

We now state that the NGFCs in these experiments were tdTom+ i.e. from the Ai14 line (**line 138**)

Lines 259-260: a correlation cannot “causally link” two observations, please rephrase.

P. 10 line 304: “causal relationship” may not be the appropriate wording

We agree with the Reviewer that this wording constitutes a potential over interpretation of the results and have removed this statement restricting our description of the data to simply stating that a correlation exists.

-The discussion could emphasize better the “cell autonomous” effect of NMDA-R hypofunction and maybe speculate a little bit on the molecular pathways linking NMDA-R hypofunction and AMPA-R development?

One could speculate on numerous cellular pathways that precipitate such changes but without extensive additional experimentation, which we feel is beyond the scope of the current manuscript (especially in light of the already large amount of data presented) this would be merely result in additional conjecture to the discussion.

Reviewer #2 (Remarks to the Author):

The study by Chittajallu R et al. tackles the role of NMDA receptors in the developmental integration of neurogliaform cells of the Stratum Lacunosum Moleculare (S-LM) in the CA1 area of hippocampus. This study comes as a follow up to previously published work by the same group on the development of Medial and Caudal Ganglionic Eminence- derived interneurons populating the CA1 area. Here, the authors tackle the role of activity through NMDA receptors in the development of the intrinsic and synaptic electrophysiological properties of a specific type of interneuron, called neurogliaform cell (NGFC). The study is timely and overall well performed by a group with significant experience in analysis of synaptic properties on and by hippocampal GABAergic interneurons. The authors report a number of alterations after removal of NMDA receptors from NGFCs during development, which are presented in a sequential and somewhat incremental manner. Although the study presents some novel findings, there are a number of studies that have assessed the role of activity in the development of cortical neurons, both pyramidal and interneurons, including neurogliaform cells, using more or less the same toolkit as the authors herein (most of them are cited in this manuscript). Importantly, the study seems to lack a core thread and has rather been laid out as a series of analyses after knocking out the *Grin1* subunit of NMDA receptors. A major re-write would facilitate in bringing up and leading to what the reviewer sees as the key message of this paper, namely the TC versus thalamic input differences between WT and KO cells.

We thank the Reviewer for his/her enthusiasm for our manuscript. We agree that although others have investigated the role of NMDAR function on interneuron development using genetic strategies our study constitutes the first extensive description of the physiological and synaptic profile across postnatal development of a hippocampal interneuron subtype under WT and *Grin1* KO conditions. This approach has allowed us to reveal that NMDARs are critically involved in the normal developmental programs regarding morpho-physiological properties and their ablation significantly alters the normal developmental progression of these parameters. Secondly, using an optogenetic approach we are the first to dissect and describe synaptic transmission onto a hippocampal interneuron subtype upon activation of extra-hippocampal inputs originating in entorhinal cortex and thalamus. Finally, we also demonstrate for the first time that NMDAR-ablation does not perturb all afferent inputs, assayed via the strength of excitatory synaptic transmission, onto an individual NGFC. Although a recent study by the Fishell Lab (Marco De Garcia et al., 2015) demonstrate that the number of presynaptic partners of cortical NGFCs in the intra-cortical versus thalamic are affected in an opposite manner after NMDAR-ablation the functional consequences of these changes were not investigated. Together, we feel that our manuscript encompasses sufficient novelty but equally as important, when taken in context with the few previous studies examining the consequences of NMDARs-ablation on interneuron function, our study clearly emphasizes that NMDA-R signaling controls numerous developmental programs that are dependent on neuronal subtype, brain region and afferent input. Nevertheless, we agree with the Reviewer that the framing of the study required a re-write and we hope that the revisions (particularly in the title/abstract/intro and results sections) made have clearly laid out what we consider the important/novel aspects of our study.

More specifically:

1. Chittajallu et al. report that NGFCs of the CA1 hippocampal area have the largest NMDA/AMPA ratio observed in any of the interneurons studied so far, the mean of which is around “3”. This is largely based on the work of the authors themselves in a previously published study (Matta J et al. 2014), in which they recorded from a number of MGE- and CGE-derived hippocampal interneurons, but not SLM ones, which include NGFCs. In that study the authors report that the highest ratios are found in CGE-derived cells, some of which have a value of “2”. Upon a closer look of the data presented in the current paper, there is a widespread distribution of ratio values ranging from 1 to 5.5 in the 16-21 age group. This means that there is an overlap between the ratios of different CGE-derived cells and the values likely form a continuum. Have the authors done a statistical comparison between the distributions of other CGE-derived cells and NGFCs to see if they are different?

In the Matta *et al.* study The CGE-derived interneurons examined were of the CCK-expressing subtype (i.e. CCK-basket, Schaffer-Collateral associated and dendrite targeting cells). We have included a figure for the reviewer that compares the NA ratios measured in these two subtypes of CGE-derived interneurons. We show that indeed the ratios are significantly different (Mann Whitney U-test; $p = 2.6 e^{-12}$) even though, as the reviewer correctly points out, that there is a wide range of NA ratios in CCK and NGFC interneuron that in some individual cases overlap in value.

2. In lines 121-123 the authors write “Therefore, Htr3A-promoter driven cre-expression is apparent from the CGE-progenitor stage but is only evident in a subpopulation of CGE-derived NGFCs demonstrating a mosaic pattern of cre-expression.” It is not clear why the authors claim that all the GFP-positive cells are NGFCs since the only recorded from them in the first set of experiments and subsequently worked on and characterized the Cre-Ai14 fate-mapped cells. Both mouse lines that the authors are using here are BAC transgenics and unless one performs double in situ hybridization analysis against the fluorophores and the actual 5-HT3A mRNA it is not easy to make such a statement. It could be, that the 5-HT3A-GFP mouse line also marks some Cajal-Retzius cells found in the SLM of CA1.

When recording from the 5HT3AR-EGFP mice and targeting reported cells in SLM are exclusively interneuron in identity. In this mouse line we do not see any reported cells that have the typical and distinctive morphological characteristics of Cajal-Retzius cells. In the double, 5HT3AREGFP/5HT3ARCre: Ai14 mouse our counts demonstrate there are a percentage of the EGFP cells that are not Ai14 positive as highlighted in **Supp. Fig. 3**. Of course it is possible that this underreporting is biased to subtypes of CGE-derived interneurons that are not NGFCs. However, we find this unlikely considering the NGFC abundance in this hippocampal lamina.

Furthermore, we have recorded from SLM cells in the double mouse and can readily identify late-spiking (n=5) NGFCs that are EGFP+ but Ai14-ve confirming our notion that the 5HT3Cre/Ai14 mouse line underreports the NGFC population.

3. In the section under “Lamina specific reduction in IN density and reelin-expression “ the authors report that the only population that they see a reduction of is the NGFCs positioned in the SLM, in striking contrast to cells that are found in other layers, in the populated Stratum Oriens (SO), but also in the less so Stratum Radiatum (SR). In light of the partial penetrance of NMDA receptor removal, have the authors performed electrophysiological recordings to test if the NMDA currents are absent in the cells in SO and SR so as to strengthen their claim about lamina specificity? The genetic tools are well chosen, although it is not clear why the authors see some cells with a complete loss of NMDA receptors and others with only partial or none. Is the “Cre” not strong enough in some cells or has it not fully turned on at the stage when the experiments were done? Have the authors ever performed a double in situ hybridization protocol to test for the expression of Cre mRNA in the GFP+ cells?

We have not performed in situ or immunocytochemistry for Cre in these mice. We target Ai14 reported cells on the assumption that this is indicative of Cre-expression in that particular cell at least at some point in its lifetime. In the assumed absence of ectopic reporter expression (we have no evidence to suggest this since Ai14 animals not crossed with our Cre lines do not demonstrate any reported cells; unpublished observation) it is possible that the sensitivity of the floxed Ai14 versus floxed Grin1 KO mice to a given amount of Cre, may differ perhaps explaining the mismatch between reporter expression and potential NMDAR knockout – this may be particularly true if Cre expression is transient for instance. We have additionally recorded from 4 SR cells (2 basket cells and 2 SCA cells) and 5 SO cells (4 identified as O-LM cells) in the 5HT3ARCre floxed Grin1 crossed mouse and demonstrate in all cells a complete penetrance of functional NMDAR mediated EPSCs (i.e. NA=0) thus the penetrance is cell-type specific. However, it is important to point out that using the same 5HT3ARCre line, at the same developmental ages where NMDAR knockout is not fully penetrant, we observe better penetrance of Gria2 knockout using AMPAR EPSC rectification as an indicator. Furthermore, in the Nkx2.1cre line full penetrance in ablation of both NMDARs and GluA2-expressing AMPARs is observed. Therefore, without further interrogation we are unclear as to the underlying reason for the penetrance issues.

Nevertheless, we have been very transparent in reporting this issue and we feel that it does not change the message of the manuscript since in many instances we parse our data to highlight the role of complete versus partial NMDAR-ablation in developmental synaptic integration of NGFCs. Furthermore, as outlined in the paper we have used the “mosaic” nature to our advantage to demonstrate cell autonomy of the alterations in synaptic development caused by NMDAR hypofunction which as Reviewers 1 and 3 point out is an important finding of the manuscript.

In conclusion this is a well-taken point and one that has also intrigued us during the course of this study. However, we feel that revealing the underlying reason for the penetrance issue in this particular 5HT3ARCre-floxed Grin1 crossed mouse would not change the overall conclusion of the manuscript.

4. It would also be helpful if the authors discussed how this partial removal may relate to the reduction in cell number the authors observe in SLM, also as compared to the results obtained by Kelsch W et al. 2014, where they show that removal of the NR2B subunit from all interneurons does not affect their numbers or distribution in the cortex.

This is a well-taken point and we anticipate that a higher penetrance strategy may result in greater deficits on cell number although this is speculative. Additionally, the Reviewer refers to a study by the Monyer laboratory in which *Grin2B* ablation has no effect on interneuron density. These counts were performed in hippocampal formation and were particular focused to those found in SO, SR, hilus and granule cell layer. Further, interrogation of PV and SOM interneurons in this study showed no deficit in these particular subtypes. In our study we did not observe any significant changes in 5HT3AR-Cre:A14 interneuron density in SR nor SO, the latter housing a significant population of 5HT3AR-expressing SOM positive interneurons. Thus, these results are in agreement with this study by Kelsch *et al.* and it appears that the role of NMDAR function in ensuring appropriate number of interneurons in the hippocampus is dependent on cell type. We thank the reviewer for this comment but we must point out that due to the overall feeling that the manuscript is rather lengthy we followed the advice of Reviewer 1 and removed the data concerning the cell number deficits. Perhaps a more thorough examination of this important developmental question employing novel strategies that are more penetrant in nature will constitute the bases for a separate study

5. The authors find that upon removal of NMDA receptors the frequency of spontaneous AMPA receptor-mediated currents increases, as does the evoked current recorded after electrically stimulating in the same layer. At the same time the dendritic length of the cells increases and the paired-pulse ratio (PPR) at P16-21 decreases. How are these findings related? And what do the authors think is the primary effect of removing NMDA receptors versus secondary?

As the reviewer is aware the sEPSC frequency is sensitive to alteration in PPR and/or increase in number of synapses that may accompany supernumerary dendritic arborization. Although not the only contributing factors influencing the measured sEPSC frequency, the question as to whether these changes are mechanistically linked is unknown. However our new analysis of dendrite complexity at P6-11 points (**Fig. 4**) to the fact that an increase in putative number of functional synapses precedes that of the changes in PPR upon NMDAR ablation and could be the precipitating factor resulting in the increase in sEPSC frequency noted at this early developmental age. Thus, these pre- and post-synaptic alterations emerge at differing developmental epochs but together at P16-21 likely both contribute to the increase in sEPSC frequency. We refer the Reviewer to the added discussion of these results in the revised text (**lines 301-332**).

6. Could the changes in the observed PPR be the result of postsynaptic rather than presynaptic changes? The authors stimulate at a fairly high frequency, 50Hz, leaving the possibility open that the synaptic depression observed in the KO is partially due to receptor desensitization. Has a lower frequency of stimulation been tried?

This is a well-taken point and concur that PPR is not solely influenced by presynaptic release properties. We have also measured the PPR at 20Hz frequency in a subset of NGFCs. The results shown in the figure are essentially similar to that seen at 50Hz presented in the manuscript. We therefore have decided to leave the 50Hz dataset in the original manuscript (**Fig. 3e**).

7. The optogenetic experiments and results are very interesting. The authors show nicely that there are terminals labeled in the SLM of CA1 after injections in either the mEC or NRe. It would be valuable if they showed how the expression looks like at the site of injection for each of the two spots.

We thank the reviewer for their enthusiasm regarding the channelrhodopsin experiments. We have now included images of the respective injection sites for the reviewer in **Supp.Fig. 6**. Furthermore, due to this enthusiasm we have also expanded our dataset regarding the optogenetic experiments to also include cortical and thalamic input onto MGE NGFCs (**Fig. 6**).

8. The authors report that only after light-stimulation of the mEC inputs, but not the NRe, in the absence of NMDA receptors they obtain a statistically-significant increase in the maximum AMPA receptor-mediated synaptic current evoked. Although there seems to be a specific increase in the mEC inputs, the variability of the evoked responses in the absence of Grin1 is quite large and with a data point that appears to be an outlier. If the authors excluded that point would they still get statistically-different responses compared to WT?

We have re-inspected the data concerning this outlier (sEPSC frequency = 13.333 Hz) and have no reason to discard the point. However, for the information of the reviewer we removed the data point and re-ran the stats and found the difference between WT and Grin1 KO to be of statistical significance (p = 0.011 versus p = 0.018; the latter p value corresponding to the statistical test in which the outlier was removed). However we need to point out that, as mentioned above we have increased our dataset regarding these experiments that now also include the addition of data from MGE NGFCs (**Fig.6**). With the greater number of n's we now demonstrate that in fact both inputs are significantly potentiated. Thus, in the revised

manuscript we conclude that extra-hippocampal inputs undergo aberrant synaptic strengthening upon NMDAR ablation.

9. Also, the reviewer was wondering how do the authors potentially relate the specificity obtained in the optogenetic experiments with the changes observed in the morphology of the cells?

This is a great question but one for which we have little insight. A detailed anatomical approach using electron microscopy would likely be required to further examine this. However these experiments are technically challenging and would be time intensive and as such are beyond the scope of the current study.

10) Methodologically, it seems that these optogenetic experiments were not performed in the presence of TTX and 4AP, as reported in previous studies (for ex. Mao T et al. Neuron 2011). Is it hence plausible that part of the current recorded does not come from the direct inputs from the mEC or NRe onto the neurogliaform cells, but rather through intermediate cells?

For the thalamic inputs we cannot envisage any origin that could result in the generation of a polysynaptic excitatory input onto NGFCs dendrites in SLM. For the optogenetic experiments in which the entorhinal cortex is infected it is clear that in addition to the temporoammonic path, the medial perforant path is clearly subject to channel rhodopsin activation (**Fig. 6b,d**). Thus, it can be envisaged that a potential polysynaptic excitatory response (via the “traditional” tri-synaptic circuitry of the hippocampus) could ultimately impinge on NGFCs, in particular the subset of NGFCs with dendrites in SR that receive Schaffer-Collateral inputs. However in these experiment we find this scenario unlikely for the following reasons

- (i) The synaptic delay of any potential polysynaptic response when compared to the direct TA input is in the order of 10-20 ms. In our experiments we never saw AMPAR-mediated EPSCs with this magnitude of delay from the onset of light stimulus (**Fig. 6e**).
- (ii) Under our conditions of stimulation (1ms) both our NMDAR and AMPAR EPSCs display similar decay kinetic signatures when compared to those elicited by local electrical stimulation (compare **Fig.1j** and **Fig. 6e**).
- (iii) Increasing our LED power to the maximum permitted (100 % arbitrary), potentially allowing for greater chance of spiking intermediate excitatory neurons, does not appreciably change the kinetics of AMPAR-mediated responses nor result in delayed polysynaptic EPSCs. In fact we routinely used this maximum LED power to evoke AMPAR-mediated EPSCs
- (iv) Eliciting trains of 5 stimuli at 50Hz that again could potentially increase the chances of spiking intermediate cells within the hippocampal network, particularly under conditions of our experiments in which picrotoxin and CGP55845A are used to block GABAergic transmission. However this does not result in traces containing obvious polysynaptic activity. Thus although the reviewer is correct to point out that we did not perform the experiments in the presence of TTX/4-AP we are confident that our light-evoked responses are essentially monosynaptic in nature.

Reviewer #3 (Remarks to the Author):

The study by Chittajallu et al. 'NMDA receptor hypofunction disrupts the developmental programs critical for appropriate circuit integration of neurogliaform cells' describes the developmental trajectory of glutamate synapse maturation in a genetically and functionally identified interneuron type in the postnatal hippocampus. The authors demonstrate with state-of-the-art methods and analyses the developmental consequences of NMDA hypofunction or loss-of-function on spontaneous synaptic inputs and presynaptic features. They exploit elegantly conditional mutant lines to generate mosaic (partial and complete) gene loss-of-function and demonstrate that the changes in synaptic development occur in a cell autonomous fashion. The paper also convincingly demonstrates that NMDAR loss-of-function differentially alters inputs to genetically-defined subsets of this interneuron type. The study thus substantially advances our view of the heterogeneity of developmental NMDAR function and will influence thinking in the field. A few points are outlined below that still need to be improved.

Major points

1. Did the authors test whether the NMDAR hypofunction observed at P16-21 would eventually switch after P21 to a bimodal distribution in dTom+ cells with either no or normal NMDAR component? In other words, is hypofunction just a transient phenomenon? This appears critical to understand the logic of the recombination event. Further, a more detailed discussion of Cre expression levels and developmental time course appear critical for understanding the results.

As indicated in the manuscript the penetrance of NR1 knockout, assessed as the distribution of NA ratios, is not different between p6-11 and p16-21 (approximately 20-25% of NGFCs have complete ablation of NMDAR function; **Fig. 1k**). Thus, at least between these two points in development no change in penetrance occurs. Regarding the Reviewer's question as to whether the NA distribution switches to a bimodal distribution as the animal ages we have now assessed NA ratios measured in NGFCs from 5HT3Cre *Grin1* KO mice between p33 and p45. These data demonstrate that, at least with the number of NGFCs tested at the latter age point, a clear bimodal distribution is not apparent in NA ratio, however, a greater penetrance of NMDA knockout as evidenced by the number of NGFCs with NA ratio of 0 is clear. This is depicted in the figure below superimposing the NA ratios in all cells between P6-11, P16-21 (already included in

Fig. 1k) and P33-45.

We feel that a detailed discussion as to the relationship between Cre-expression levels during development and NMDAR ablation would be superfluous in so far as it would not change the interpretation of our results. As mentioned above (Response 3 to Reviewer 2) we have been very transparent in reporting the penetrance issues. In fact, we have used the “mosaic” nature to our advantage to demonstrate cell autonomy of the alterations in synaptic development caused by NMDAR hypofunction that, as both yourself and Reviewer 1 point out, is an interesting and important finding of our study.

2. At which time point is the reduced number of dTom+ cells in GluN1 ko mice first observed (related to Fig. 2a,b,g). Here further evidence needs to be provided whether cells are lost during maturation after they took their eventual position and/or already display impaired migration.

Due to the large amount of data present in the manuscript and in response to Reviewer 1 we have chosen to remove our cell counting data from the revised version. However, we concur that further evaluation of the underlying cellular mechanisms (i.e. cell death versus migration) would be extremely interesting but one which would be suited for a separate study.

3. Were reconstructions made from 50 μm thick slices as described in the histology section? Please provide here further technical details. If indeed 50 μm sections were used and considering the indicated span of dendrites $>200 \mu\text{m}$, truncation is expected to be substantial. Changes in morphological feature of total dendritic length vs branching may be hard to assess. Also, it is not clear in how far synapse densities correlates to dendrite surface. Considering these points, the authors may moderate the conclusions and add Fig. 5 e-l as supporting evidence to the Suppl. Mat. or provide stronger evidence for correlations of structural synapse densities and dendritic length in wt and mutant cells. The same question essentially applies to Fig. 8e-l where the larger SLM dendrites are more prone to the above mentioned analysis problems.

As the review points out one would expect the dendrites to traverse the z-axis to a similar extent to that seen in the x-y axes (i.e. approximately $100 \mu\text{m}$ away radially from the soma in all directions). The re-sectioning after biocytin staining was in fact done at a thickness of $70 \mu\text{m}$ and not at the $50 \mu\text{m}$ written in the manuscript. We apologize for the error and have amended the methodology section accordingly. However, even with this additional $20 \mu\text{m}$ we do agree completely with the possible limitations of structural analysis in re-sectioned slices. However, we must point out that we only analyzed NGFCs that possessed dendrites in one re-sectioned slice only- in all cases the soma of these NGFCs were found in the uppermost re-sectioned slice (i.e. the face of the slice from which the recording electrode approaches). Thus, we are confident of a full re-construction of the dendrites that project “below” the soma. Of course it is possible that we have severed portions of dendrites projecting “above” the slice from which the recording was performed. Nevertheless, we feel that this would “work against us” in terms of the observing supernumerary dendrite intersections. Furthermore, our Sholl analyses clearly indicate that this exuberance is limited to the “central portion” of the dendritic arbor (typically between $40 - 100 \mu\text{m}$ from the soma center). Finally, for these reasons we also examined the existence of correlations between sEPSC frequency and dendrite complexity and these analyses clearly demonstrate a relationship between these morphological and synaptic functional parameters (**Fig. 4h, and Supp. Fig. 7**). As a consequence this gives added confidence

regarding our conclusions based on the Sholl analyses performed. Nevertheless, we understand the need to be cautious here and toned down the description of these data. Of course, as the reviewer points out, a more direct anatomical analysis are necessary to definitively implicate changes of synapse number in the observed functional abnormalities observed in the current study. These additional considerations/caveats are now explicitly stated in the revised manuscript text (**lines 320 –327**)

4. Discussion: Except for the detailed difference between the present study and the two previous studies on NMDAR deletion in INs, the ms. would benefit from a more focused and critical discussion of their own findings e.g. cell autonomous effects and potential relation of the different changes in synapse function. Also, a more careful discussion of NGFCs and their relation to other interneuron types in disease context would strengthen the ms. In particular, evidence of dendrite vs. soma-targeting INs in pathology may be disentangled as far as possible.

We agree with the need to revise sections of the manuscript to incorporate some of the points mentioned here and have done so in the revised manuscript. In particular, with regard to the potential relation of the different changes in synapse function we have now included a section within the results (“Synaptic underpinnings of the NMDAR hypofunction-mediated abnormal exuberance in sEPSC frequency”) to address this point.

Minor points

1. Please define MGE/CGE when first used.

We thank the reviewer for pointing out this omission and the text has been amended accordingly.

2. Please provide technical details how cells were quantified in Suppl. Fig 1. Please check whether ‘hippocampi’ actually should say ‘section’ in the Fig. legend: ‘8-10 hippocampi counted per mouse; number of cells counted per hippocampus = 7 - 21). Was the entire SLM imaged or only part of it in each section?

We apologize for the error here. The sentence should in fact read “8-10 hippocampal sections counted per mouse” and have changed the text accordingly. Furthermore, we have specified that the entire SLM was analyzed for the cell counts. Also please note that in the revised manuscript this is now **Supp. Fig. 3**.

3. Green lines are hard to distinguish in Fig. 1o.

We have made the lines thicker for easier visualization in the new **Fig. 1k**

4. Related to Fig. 3d, p-values should be provided in the text for the comparison between WT vs. Cre- in grin1 ko and NA<1 vs Cre- in grin1 ko cells at P16-20 to support conclusions.

The statistical significances between the datasets mentioned are now included within the figure panel itself. Note that this is now **Fig. 2e** in the revised manuscript.

5. The 'causal' relationship is well demonstrated between the loss of GluN1 and PPR or EPSC freq changes, respect. in a cell autonomous fashion, but are PPR changes are necessarily 'causal' to sEPSC freq changes as claimed in l. 260? Or could two effects co-exist and for instance (functional) synapse densities differ additionally between mutant and wt cells?

We agree with the Reviewer and in fact similar concerns were raised by Reviewer 1. This wording constitutes a potential over interpretation of the results and have we have therefore removed this statement restricting our description of the data to simply stating that a correlation exists. In the revised manuscript as a response to Reviewer 1 (point 4) we have now included Sholl analyses at the P6-11 point (**Fig. 4**). The results of these data allows us to make certain conclusions regarding the relationship between PPR and dendrite complexity (as a proxy for synapse number/density) in precipitating the increased sEPSC frequency observed during development (again we kindly refer the Reviewer to the Results section "Synaptic underpinnings of the NMDAR hypofunction-mediated abnormal exuberance in sEPSC frequency").

REVIEWERS' COMMENTS:

Reviewer #1 (Remarks to the Author):

This revised version of the manuscript is now much more focused and easy to read. The authors have addressed all the comments I had raised (although I must say that a couple of sentences in the rebuttal letter were partly truncated!); I therefore recommend this article for publication.

Reviewer #2 (Remarks to the Author):

The authors have addressed the concerns and questions I raised at a sufficient degree. They have made the manuscript tighter and the message more concise by removing part of the data and re-writing the text.

Reviewer #3 (Remarks to the Author):

The authors addressed the main points raised. The study is now well focused and will be of great interest to the broad readership of Nature Communications.